# Divergence in a eukaryotic transcription factor's co-TF dependence involves multiple intrinsically disordered regions

Lindsey F. Snyder[1], Emily M. O'Brien[2], Jia Zhao[2,11], Jinye Liang [2], Baylee J. Bruce[1], Yuning Zhang[3,12], Wei Zhu[4], Thomas H. Cassier[2], Nicholas J. Schnicker [5,6], Xu Zhou [7], Raluca Gordân[3,4,8,9,10] & Bin Z. He [1,2] ✉

Combinatorial control by transcription factors (TFs) is central to eukaryotic gene regulation, yet its mechanism, evolution, and regulatory impact are not well understood. Here we use natural variation in the yeast phosphate starvation (PHO) response to examine the genetic basis and species variation in TF interdependence. In *Saccharomyces cerevisiae*, the main TF Pho4 relies on the co-TF Pho2 to regulate ~28 genes, whereas in the related pathogen *Candida glabrata*, Pho4 has reduced Pho2 dependence and regulates ~70 genes. We found *C. glabrata* Pho4 (CgPho4) binds the same motif with 3–4 fold higher affinity. Machine learning and yeast one-hybrid assay identify two intrinsically disordered regions (IDRs) in CgPho4 that boost its activation domain's activity. In ScPho4, an IDR next to the DNA binding domain both allows for enhanced activity with Pho2 and inhibits activity without Pho2. This study reveals how IDR divergence drives TF interdependence evolution by influencing activation potential and autoinhibition.

Transcription factors (TFs) are the cornerstone of gene regulatory networks. In eukaryotes, TFs often work collaboratively to regulate gene expression. This combinatorial control is crucial for enhancing specificity, because most eukaryotic TFs recognize short and degenerate motifs—typically less than 10 bps—that appear hundreds to tens of thousands of times in the genome[1]. TFs with the same family of DNA binding domains (DBDs) also recognize highly similar motifs[2]. Despite these, TFs often bind only a fraction of their motifs in vivo and regulate an even smaller subset of the genes they bind to[3]; paralogous TFs regulate distinct sets of genes[4,5]. The key to achieving this specific regulation is the requirement of two or more TFs to jointly regulate the target genes[6–8]. Another important function of combinatorial control and TF interdependence is to allow cells to integrate multiple upstream signals. In fly development, for example, a combination of tissue-specific "selector" and morphogen gradient—both are TFs—precisely determine the expression pattern and define the cell fate[9].

The importance of combinatorial control and the molecular mechanisms behind TF interdependence are traditionally studied using mutations that disrupt TF interactions. For instance, a missense mutation that disrupts the interaction between two cardiac TFs, GATA4 and TBX5, was shown to cause dysregulation of cardiac genes and lead to malformation of heart tissues[10]. Similarly, studies of

[1]Interdisciplinary Graduate Program in Genetics, University of Iowa, Iowa City, IA, USA. [2]Department of Biology, University of Iowa, Iowa City, IA, USA. [3]Department of Biostatistics & Bioinformatics, Duke University, Durham, NC, USA. [4]Department of Molecular Genetics & Microbiology, Duke University, Durham, NC, USA. [5]Protein and Crystallography Facility, University of Iowa, Iowa City, IA, USA. [6]Department of Molecular Physiology and Biophysics, University of Iowa, Iowa City, IA, USA. [7]Department of Pediatrics, Division of Gastroenterology, Hepatology and Nutrition, Boston Children's Hospital and Harvard Medical School, Boston, MA, USA. [8]Department of Computer Science, Duke University, Durham, NC, USA. [9]Department of Cell Biology, Duke University, Durham, NC, USA. [10]Department of Genomics and Computational Biology, University of Massachusetts Chan Medical School, Worcester, MA, USA. [11]Present address: Laboratory of Immunophysiology, The Ragon Institute of Mass General, MIT, and Harvard; Department of Biology, Massachusetts Institute of Technology, Cambridge, MA, USA. [12]Present address: Department of Genetics, Washington University School of Medicine, St. Louis, MO, USA. ✉e-mail: bin-he@uiowa.edu

combinatorial control evolution often focus on losses (and gains) of TF interactions, like the MADS-box TF, Mcm1, and its co-TFs, MATa2 and MATα2 in the yeast mating type pathway[11,12]. A far less explored type of change in the combinatorial control is a change in the TF interdependence itself, that is, when TFs involved in cooperative regulation evolved to be more or less dependent on the co-TF(s). Such changes are expected to dramatically rewire the network by altering the specificity and signals required for gene regulation, which could have profound implications in disease and evolution. In relation to this possibility, not all eukaryotic TFs require other TFs to function. The yeast TF, Gal4, regulates its target genes on its own[13]. This raises several intriguing and unanswered questions: if TF interdependence and combinatorial control evolve, what genetic changes underlie the divergence, what aspects of the TF activities are impacted, and what are the consequences on the regulatory output?

Intriguingly, natural variation in co-TF dependence exists in the yeast phosphate starvation (PHO) response network. In the model yeast *Saccharomyces cerevisiae*, the main TF of the PHO response, Pho4 (hereinafter as ScPho4), strongly depends on the co-TF, Pho2, to induce 27/28 of its target genes[3]. In a related human yeast pathogen, *Candida glabrata*, its Pho4 (hereinafter as CgPho4) is far less dependent on Pho2 and induces twice as many genes (Fig. 1A)[14,15]. The level of Pho2-dependence varies quantitatively among Pho4 orthologs in other yeasts and is correlated with the number of genes induced in a common genome background[15]. This latter observation is consistent with the role of combinatorial control in enhancing specificity. What remains unknown is what genetic differences between Pho4 orthologs underlie the divergence in co-TF dependence, and what TF activities are impacted by those variations.

Here, we propose two non-mutually exclusive models for the difference in Pho2-dependence between ScPho4 and CgPho4 (Fig. 1B). The first "enhanced activity" model is based on studies showing that ScPho4 requires Pho2 to (1) bind cooperatively to the target gene promoters[16], (2) recruit the histone acetyltransferase (HAT) complex[17] and (3) help recruit general TFs and the PolII complex[18]. Under this model, we hypothesize that CgPho4 binds more tightly to DNA than ScPho4, and is more capable of recruiting general transcription factors (TFs) and the PolII complex, thereby becoming less Pho2-dependent. The second "autoinhibition" model is based on a study suggesting that ScPho4 is auto-inhibited, and that interaction with Pho2 unmasks its activation domain and allows it to function[19]. This model predicts that CgPho4 either lacks or has far weaker effects of the auto-inhibition and hence doesn't depend on Pho2.

In this study, we tested these two models by comparing the DNA binding and activation abilities of the two Pho4 orthologs, and systematically swapping regions between the two Pho4s in a series of 50 chimeric TFs, then quantifying their activities with and without Pho2. Our results support both models as contributing to the Pho2-dependence variation. To our surprise, while CgPho4 DBD binds to the same consensus motif with a 3–4-fold higher affinity than ScPho4 DBD, swapping DBD alone failed to yield the expected results. Instead, the differences in Pho2-dependence originated primarily from differences in the Intrinsically Disordered Regions (IDRs) in the two TFs that modulate both their Activation Domain (AD) and DNA binding domain (DBD) activities. Therefore, our results reveal that evolution in a eukaryotic TF protein, particularly through changes in the IDR, can lead to divergence in co-TF dependence, which in turn results in a more than two-fold change in the size of the target network.

## Results

### Domain organization and sequence divergence between ScPho4 and CgPho4

Based on genetic and biochemical studies, ScPho4 encodes the following functional domains from its N- to its C-terminus (Fig. 1C): a regulatory region (R1, aa 1–42) interacting with the negative regulator Pho80[20], the activation domain (AD, aa 43–99)[21,22], a region encoding the nuclear export and import signals (NLS, aa 100–176)[23,24], a protein-interaction domain interacting with both Pho80 and the co-TF, Pho2 (referred together as P2ID, aa 177–242)[20,25], and the bHLH DNA binding domain (DBD, aa 243–312)[21,26]. Regarding ScPho4-ScPho2 interaction, a previous study mapped the region in ScPho4 required for the interaction to aa 200–218[25]. The same 18 aa stretch also contains the phosphorylation site that directly controls the TF-TF interaction[23].

Both ScPho4 and CgPho4 were predicted to be mostly intrinsically disordered outside the DBD, with some low confidence helices in R1 and AD (Supplementary Fig. 1). CgPho4 is significantly longer than ScPho4 (533 vs 312 amino acids, Fig. 1C). The NLS and P2ID regions contain four functionally important phosphorylation sites targeted by the cyclin-dependent kinase complex Pho80/85[24,27]. Those located in the NLS control Pho4's nuclear translocation, while phosphorylation of the one in P2ID disables ScPho4 from interacting with Pho2[23]. All five Pho80/85 targeted phosphorylation sites are clearly identifiable in the alignment (Supplementary Fig. 2); in fact, they were previously found to be conserved in orthologs outside the Saccharomycotina subphylum[28]. Thus, while the protein length and sequence diverged significantly between the two Pho4 orthologs, the domain architecture and post-translational modification motifs appear to be highly conserved in evolution.

### CgPho4 binds the same consensus motif with a higher affinity compared to ScPho4

To test the enhanced activity model, we first compared the DNA binding ability of the two Pho4 orthologs. First, we asked if the two Pho4 orthologs recognize the same DNA sequence. The N-terminal stretch of the first α-helix in the bHLH domain, known as the basic region, contains the residues determining sequence specificity. All four residues known to recognize the nucleotide bases and five additional residues that recognize the phosphate backbone in ScPho4[26] are all conserved in CgPho4 except for an R252K change (number based on ScPho4, Fig. 2A). Next, we used our published Chromatin-Immunoprecipitation (ChIP-seq) data for both Pho4 orthologs to identify their respective motifs[15] (Materials and Methods). Consistent with the amino acid sequence conservation, the results showed that they bind the same E-box motif "CACGTG", with no obvious differences in the flanking nucleotides (Fig. 2B). These motifs were based on a relatively small number of ChIP peaks (74 for ScPho4 and 118 for CgPho4). To comprehensively characterize and compare the binding preferences of the two proteins, we applied Protein Binding Microarray (PBM) to map the binding landscape in the entire 7-mer space (universal PBM, Fig. 2C). The result revealed a lack of divergence in their sequence preference (Pearson's r = 0.89). To complement the short oligo length in the universal PBM and further examine differences in the flanking base preference, we designed a second genome context library, which includes 36-bp sequences centered on ChIP-identified binding sites for both Pho4's in their native genome along with the flanking nucleotides[29]. Similar to the uPBM, the genome context PBM revealed no evidence for binding specificity differences between ScPho4 and CgPho4 (Supplementary Fig. 3, Spearman's ρ = 0.84 and 0.86 for sequences containing the consensus E-box or non-consensus variants, respectively, Materials and Methods). We conclude that ScPho4 and CgPho4 have the same sequence specificity.

Next, we tested the hypothesis that CgPho4 binds more tightly than ScPho4 does. To do so, we purified the DBD region of both Pho4s (Materials and Methods) and measured their binding affinity to a 17-bp oligo based on the *S. cerevisiae PHO5* promoter with the consensus E-box motif "CACGTG". Biolayer Interferometry (BLI) measurements showed that CgPho4 DBD binds the consensus motif >3 times more tightly than ScPho4 DBD does ($K_D$ = 5.2 nM vs 1.2 nM for ScPho4 and CgPho4, respectively; Student's t-test for log $K_D$ difference $P < 0.01$, Fig. 2D, E). We confirmed this $K_D$ difference using Electrophoretic

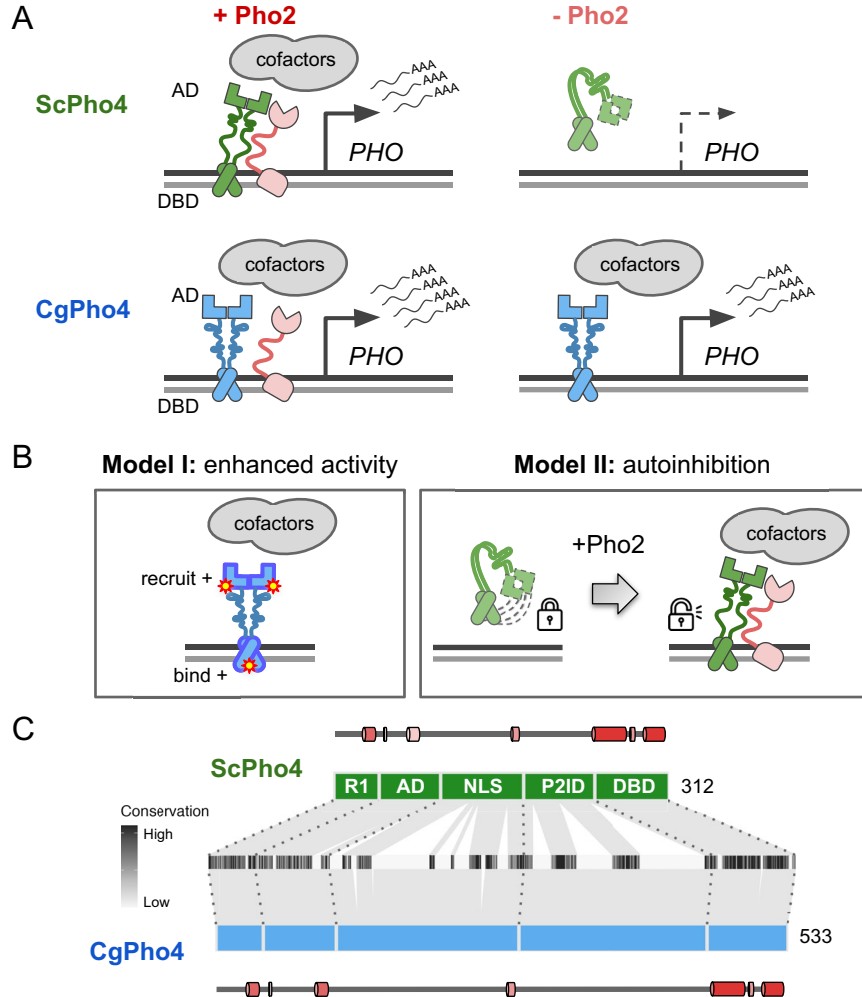

**Fig. 1 | Different co-TF (Pho2) dependence between orthologous TF (Pho4) in yeast species. A** Cartoon summary of previous work: *S. cerevisiae* Pho4 (ScPho4) depends on the co-TF, Pho2, for DNA binding and activation of Phosphate Starvation (PHO) Response genes (top row), while Pho4 from *C. glabrata* (CgPho4) can bind and activate without the co-TF. See text for references. **B** Two non-mutually exclusive models to explain the divergence in Pho2-dependence. **C** The tiles in the middle show the conservation scores at aligned positions of the two Pho4 orthologs; the box diagrams represent ScPho4 and CgPho4, with gray lines connecting matching residues to the alignment tiles. Five regions were delineated based on knowledge about ScPho4: R1 = regulatory, interact with Pho80; AD activation domain, NLS nuclear import and export signals, P2ID Pho2-interaction domain (also includes a region contacting Pho80), DBD basic Helix-Loop-Helix DNA binding domain. The lines and cylinders above and below the region diagrams show predicted α helices in each protein, with lighter color corresponding to lower prediction confidence. Prediction was done using PSIPRED 4.0.

Mobility Shift Assay ($K_D$ = 5.5 nM vs 1.9 nM for ScPho4 and CgPho4, respectively; Supplementary Fig. 4). We conclude that CgPho4 recognizes the same E-box motif and binds with a higher affinity than ScPho4. It is worth noting that this observed difference in affinity, while statistically significant, is modest ($\Delta\Delta G$ = −0.87 kcal/mol, based on the mean BLI measurements, assuming a temperature of 25 °C). Its potential impact on gene regulation depends critically on the effective concentration of Pho4 in the nucleus.

## CgPho4 encodes two additional activation enhancer domains (AEDs)

We next tested the hypothesis that CgPho4 has increased trans-activation potential compared to ScPho4 under the enhanced activity model. We first predicted regions with activation potential in both Pho4 orthologs using PADDLE, a Convolutional Neural Network trained on 150 activation domains (ADs) from 164 TFs in *S. cerevisiae*[30]. PADDLE recovered the experimentally identified AD in ScPho4 between aa 60–102, which contained the 9aaTAD motif previously described as the minimum residues required for

activation[31] (Fig. 3A, orange triangle). No other regions with significant activation potential were predicted in ScPho4. By contrast, three regions in CgPho4 were predicted to have activation capabilities, including one corresponding to the AD in ScPho4, which was predicted to have strong activity (Z-score > 6, Fig. 3B). Two additional regions, one overlapping R1 and the other spanning the NLS and P2ID, were predicted to have medium activation strength (Z-score > 4). We refer to these two regions as E1 and E2, respectively from here on. Interestingly, the best match to the 9aaTAD motif pattern in CgPho4 was found in E2 rather than in the AD (Fig. 3B, orange triangle).

To determine the activity of the predicted regions with activation potential in both Pho4s, we set up a yeast one-hybrid (Y1H) assay, in which each candidate region was fused to the Gal4 DBD and its activation potential was measured by a genome-integrated *GAL1*pr-mCherry reporter in *S. cerevisiae* (Fig. 3C, Materials and Methods). To see if fusing the candidate region with Gal4 DBD created a new activation domain, we applied PADDLE to all constructs and observed no peaks either in the Gal4 DBD or in the connecting regions

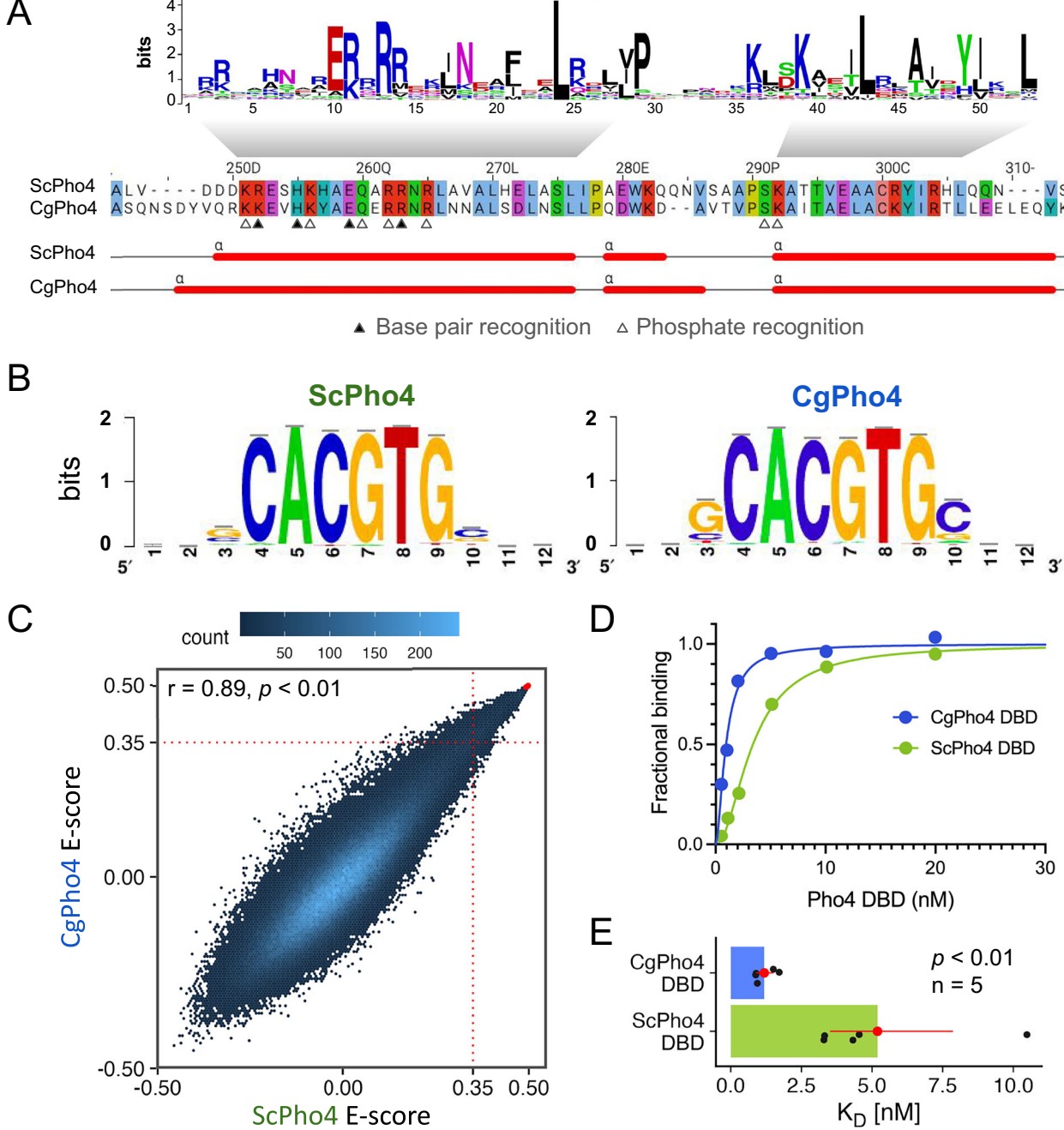

**Fig. 2 | CgPho4 DBD recognizes the same motif as ScPho4 DBD and has a higher affinity towards the consensus DNA. A** Alignment of the DBD region between the two Pho4 orthologs. Top: sequence logo for the bHLH domain (PROSITE PS50888). Middle: coordinates above are based on ScPho4. Filled and open triangles indicate residues in ScPho4 DBD making either base or phosphate backbone contacts (Shimizu et al.[26]). Bottom: predicted α-helices in both Pho4 DBDs. **B** WebLogo motifs for ScPho4 and CgPho4 are derived from ChIP-seq peaks. **C** Universal Protein Binding Microarray (uPBM) comparing the binding of full length ScPho4 and CgPho4 against all possible 9-mers. A rank-order based E-score is plotted for all 7-mers, where a score greater than 0.35 indicates specific TF-DNA binding (Gordân et al.[29]). Red dots in the top right corner represent oligos with the "CACGTG" motif. Pearson's correlation coefficient and a $p$-value based on the t-distribution are shown at the top. **D** Binding to the 17 bp oligo containing a consensus E-box motif is measured by Biolayer interferometry (BLI). Fractional binding is plotted vs the concentration of purified ScPho4 or CgPho4 DBD. One of the representative binding curves is shown here. **E** Dissociation constants ($K_D$) are determined from the binding curves and shown as dots (n = 5 biological replicates). The bars and the red dots are the means and the red lines the 95% confidence intervals from bootstrapping. Two-sided Student's t-test was performed on the log transformed $K_D$. A Wilcoxon rank-sum test on raw $K_D$ similarly yielded $P < 0.01$.

(Supplementary Fig. 5). The yeast one-hybrid result confirmed that both ScPho4 and CgPho4 ADs were able to activate the reporter above the background level (7- and 5.6-fold, Holm-Bonferroni corrected $P < 0.01$). We were able to further localize the required residues for activation in ScPho4's AD to a region of 32 aa centered on the 9aaTAD

motif (Fig. 3A, C). Neither E1 nor E2 from either Pho4 activated the reporter on its own (Fig. 3C). This was surprising for CgPho4's E1 and E2, since both had a medium strength Z-score prediction, and the latter also contained a match to the 9aaTAD (Fig. 3B). We hypothesized that E1 and E2 in CgPho4 could enhance the activity of the main AD. To test

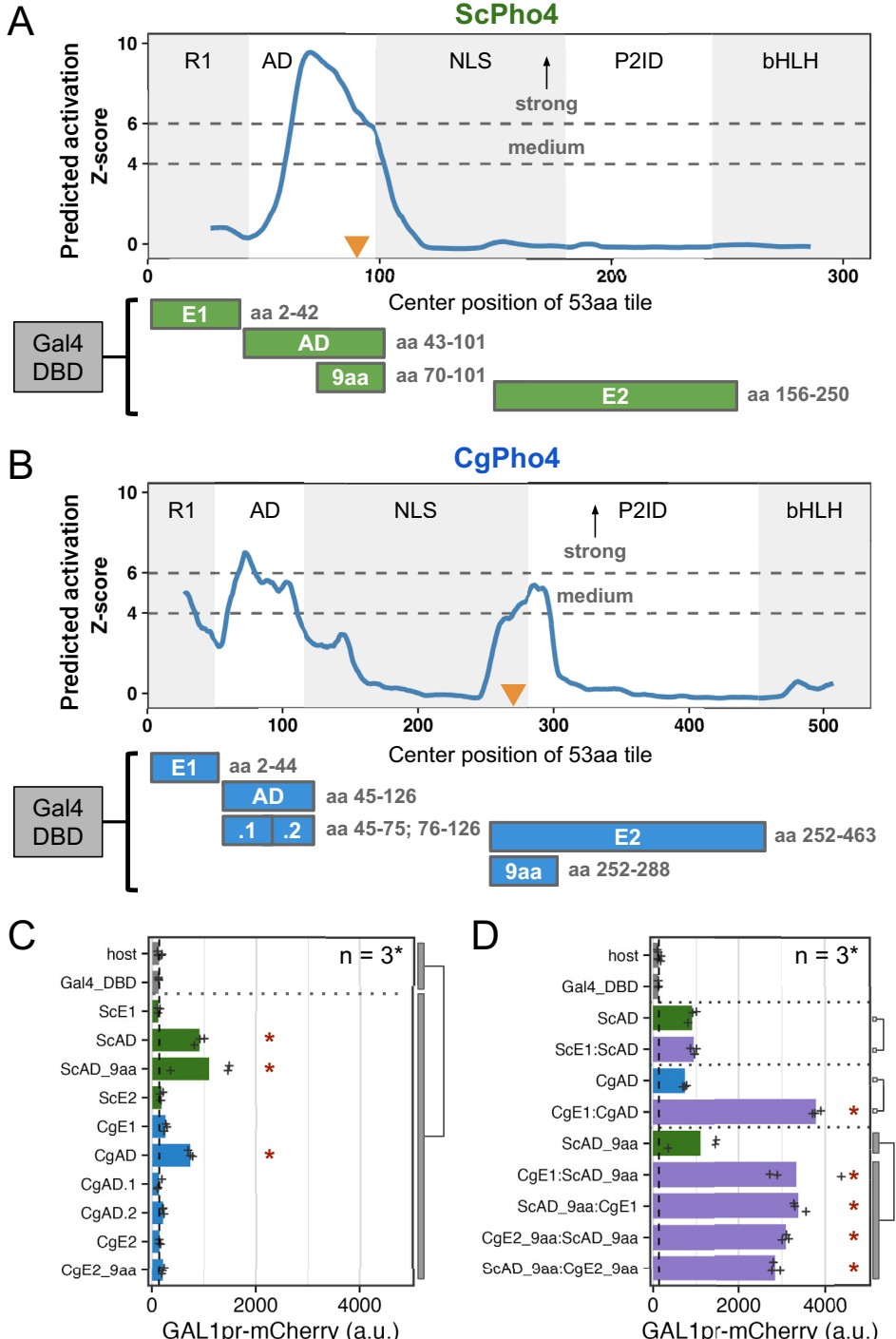

**Fig. 3 | CgPho4 encodes two Activation Enhancer Domains (AEDs) that can enhance the activation potential of the main AD. A**, **B** PADDLE prediction of activation potential in ScPho4 and CgPho4. Predictions were made in 53 aa blocks and plotted using the middle of each block as the x coordinate. As a result, the first and last 26 aa positions have no scores. Z-scores > 4 and 6 indicate regions with medium and strong activation potential, respectively. The five regions in each protein are labeled as in Fig. 1. Orange triangles mark the 9aaTAD motif matches. Below the plots are diagrams of the Gal4DBD fusion constructs used to test the activation potential, with their boundaries labeled on the right. **C** Activation potential of each region was tested in the context of a Gal4 DBD fusion driving a *GAL1pr*-mCherry reporter. Shown are the mean (bar) and individual data points

(n = 3, except for the host, for which n = 6, biological replicates). Host has the mCherry reporter but no plasmid; Gal4_DBD has the reporter and the Gal4 DBD alone. A linear model with construct as the independent variable was used to compare the Gal4 DBD fusions to the two negative controls combined; an asterisk indicates significant difference at a 0.05 level after Holm-Bonferroni correction. **D** CgE1 and CgE2_9aa can enhance the activation capability of ScAD_9aa in an orientation-independent manner. Plot design and replicate numbers are the same as in (**C**). An asterisk indicates significance difference at a 0.05 level by an unpaired, two-sided Student's t-test between the indicated groups after Holm-Bonferroni correction.

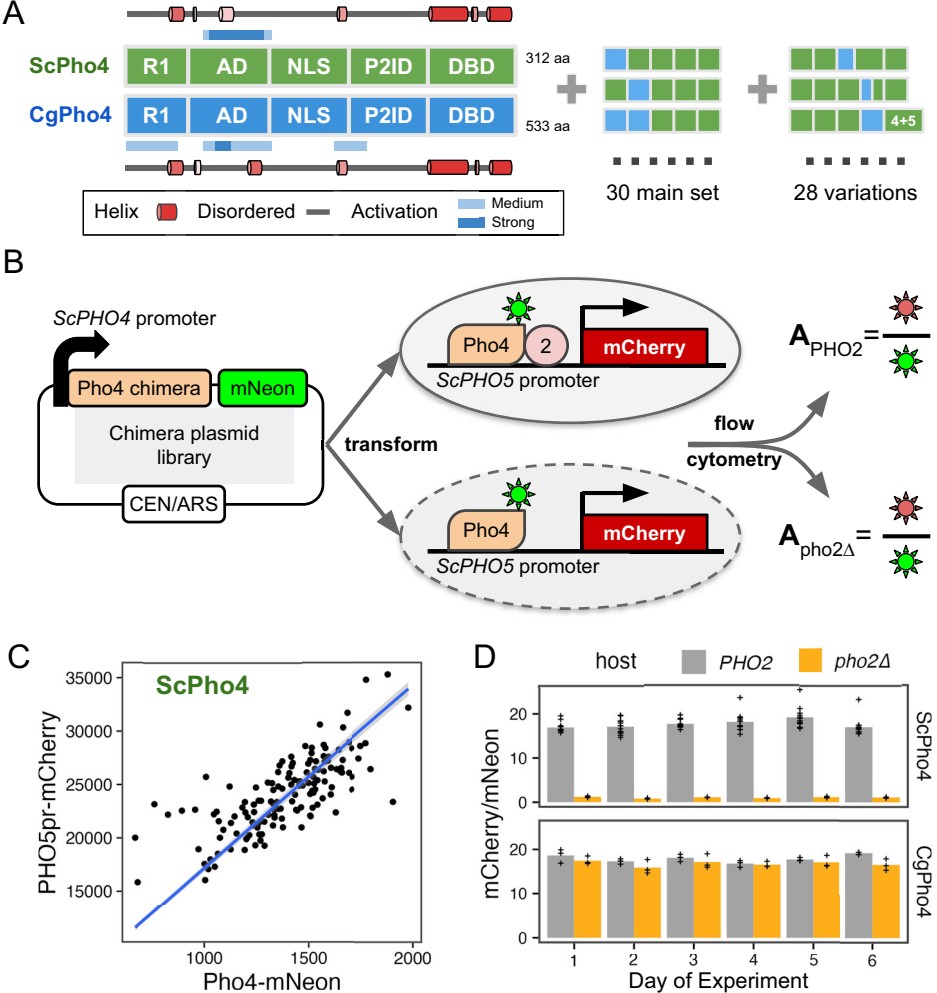

**Fig. 4 | A dual fluorescence reporter system for accurately quantifying the activity of Pho4 chimeras with and without Pho2. A** ScPho4 and CgPho4 were divided into five corresponding parts. Names of each region were based on the annotations for ScPho4. Predicted alpha helices were drawn as cartoons outside of the box diagrams. All $2^5 = 32$ combinations (2 were the original proteins) were made in the main chimera set, and an additional set of chimeras were constructed to test specific hypotheses. **B** A diagram showing the design of the dual fluorescence reporter system. The chimeric Pho4 fragments were cloned downstream of the endogenous *ScPHO4* promoter and were tagged at the C-terminus with an in-frame mNeon fluorescent protein, followed by the endogenous *ScPHO4* 3'UTR and terminator. The plasmid contains a CEN/ARS element, resulting in stable inheritance and low biological variation. **C** The *PHO5pr-mCherry* levels correlate with the ScPho4-mNeon levels between biological replicates. A 95% confidence band centered on the linear regression line in blue is shown as a gray ribbon. **D** $A_{PHO2}$ and $A_{pho2\Delta}$ values for ScPho4 and CgPho4 across 6 different days of experiments showing consistent results (n = 3 for all groups except ScPho4 with Pho2, where n = 12, biological replicates, separate inoculates at the overnight growth stage.).

this, we first fused ScE1 and CgE1 to their respective AD, and found that only the latter resulted in a significant enhancement in the activation potential (Fig. 3D). Next, we fused CgE1 or CgE2 in both orientations to the minimal AD of ScPho4 (ScAD_9aa). We found that, indeed, both regions were able to significantly enhance the activity of the ScAD_9aa in an orientation-independent manner (Fig. 3D). It is worth noting that this enhancement activity does not involve the endogenous ScPho4 or ScPho2. ScPho4 is inactive in the phosphate replete condition in which the Y1H experiment was conducted. To test the involvement of ScPho2, we repeated the assay in a host lacking *pho2*. No significant difference in the boosting effects of either CgE1 or CgE2 was observed, although CgE1's effect is slightly lower without Pho2 (Supplementary Fig. 6).

In summary, we found that CgPho4 encodes two Activation Enhancer Domains (AEDs), both of which are in the Intrinsically Disordered Region (IDR). The AEDs have little activation capability on their own but can significantly enhance the activation potential of the AD of either of the Pho4 orthologs. Since these two AEDs are present in

CgPho4 but not in ScPho4, we hypothesize that they contribute to CgPho4 being less dependent on Pho2.

## A dual fluorescence reporter assay accurately measures the activity of Pho4 chimeras with and without Pho2 for dissecting the divergence in co-TF dependence

So far, we compared DNA binding and activation potential between CgPho4 and ScPho4 by isolating individual regions and testing them either in vitro or in a synthetic in vivo system. While they support the enhanced activity model, it remains unclear whether and how they contribute to divergence in Pho2-dependence in the native Pho4 protein context. They also do not test or rule out the autoinhibition model, which necessitates intramolecular interactions. To answer these questions, we divided ScPho4 and CgPho4 into five corresponding parts (Figs. 1C, 4A), with breakpoints chosen to be on the edge of well-aligned regions to avoid breaking known or predicted functional domains and secondary structures. Boundaries for all regions used in this and other experiments can be found in Table 1. We

**Table 1 | Breakpoints for chimeric Pho4 and individual regions tested in this study**

| Origin | Region Name | Start | End | Length | Used in figure(s) |
|--------|-------------|-------|-----|--------|-------------------|
| CgPho4 | R1 | 1 | 44 | 44 | Figs. 1, 4, 5, 6, S2, S4 |
| CgPho4 | AD | 45 | 112 | 68 | Figs. 1, 4, 5, 6, S2, S4 |
| CgPho4 | NLS | 113 | 282 | 170 | Figs. 1, 4, 5, 6, S2, S4 |
| CgPho4 | P2ID | 283 | 458 | 176 | Figs. 1, 4, 5, 6, S2, S4 |
| CgPho4 | DBD | 459 | 533 | 75 | Figs. 1, 4, 5, 6, S2, S4 |
| ScPho4 | R1 | 1 | 42 | 42 | Figs. 1, 4, 5, 6, S2, S4 |
| ScPho4 | AD | 43 | 99 | 57 | Figs. 1, 4, 5, 6, S2, S4 |
| ScPho4 | NLS | 100 | 176 | 77 | Figs. 1, 4, 5, 6, S2, S4 |
| ScPho4 | P2ID | 177 | 242 | 66 | Figs. 1, 4, 5, 6, S2, S4 |
| ScPho4 | DBD | 243 | 312 | 70 | Figs. 1, 4, 5, 6, S2, S4 |
| CgPho4 | P2ID first half | 283 | 327 | 45 | Figs. 6, S2 |
| CgPho4 | P2ID second half | 328 | 458 | 131 | Figs. 6, S2 |
| ScPho4 | P2ID first half | 177 | 204 | 28 | Figs. 6, S2 |
| ScPho4 | P2ID second half | 205 | 242 | 38 | Figs. 6, S2 |
| CgPho4 | E1 | 2 | 44 | 43 | Figs. 3, S5 |
| CgPho4 | AD | 45 | 126 | 82 | Figs. 3, S5 |
| CgPho4 | E2 | 252 | 463 | 212 | Figs. 3, S5 |
| CgPho4 | AD.1 | 45 | 75 | 31 | Figs. 3, S5 |
| CgPho4 | AD.2 | 76 | 126 | 51 | Figs. 3, S5 |
| CgPho4 | E2_9aa | 252 | 288 | 37 | Figs. 3, S5 |
| ScPho4 | E1 | 2 | 42 | 41 | Figs. 3, S5 |
| ScPho4 | AD | 43 | 101 | 59 | Figs. 3, S5 |
| ScPho4 | AD_9aa | 70 | 101 | 32 | Figs. 3, S5 |
| ScPho4 | E2 | 156 | 250 | 95 | Figs. 3, S5 |
| ScPho2 | AD | 404 | 559 | 156 | Fig. 3 |
| ScPho4 | Alternative NLS | 100 | 156 | 57 | Fig. 5 |
| CgPho4 | Alternative P2ID | 253 | 463 | 211 | Fig. 5 |
| ScPho4 | ΔbHLH | 2 | 250 | 249 | Fig. 6 Y2H |
| CgPho4 | ΔbHLH | 2 | 463 | 462 | Fig.6 Y2H |
| ScPho2 | Pho4int | 112 | 407 | 296 | Fig. 6 Y2H |
| CgPho4 | purified DBD | 452 | 533 | 82 | Figs. 2, S4 |
| ScPho4 | purified DBD | 236 | 312 | 77 | Figs. 2, S4 |

created all $2^5 = 32$ combinations of these regions as well as 28 additional ones with alternative breakpoints (Fig. 4A). All constructs were C-terminally tagged with mNeon to quantify the protein levels, and were expressed from the native *ScPHO4* promoter and UTRs (Fig. 4B). We also created two *S. cerevisiae* host strains, in which the *ScPHO5* CDS was replaced by an mCherry reporter, and *pho80* was knocked out so that all Pho4 chimeras were constitutively nuclear localized[24,27]. One of the two host strains had *pho2* knocked out. For each chimera, we measured its *PHO5pr*-mCherry and Pho4-mNeon levels using flow cytometry in both hosts. While mCherry and mNeon levels for the same Pho4 construct varied between experiments, the ratios between the two were consistent and characteristic of the specific construct (Fig. 4C, D). We therefore defined the activity of a Pho4 chimera as the ratio between the median fluorescence intensity (MFI) of mCherry and mNeon, which we will refer to as $A_{PHO2}$ or $A_{pho2\Delta}$ from hereon.

### The two activation enhancer domains (AEDs) in CgPho4 increased the activity of the chimeric TFs but were insufficient to make them Pho2-independent

The chimeric Pho4 constructs exhibited varied $A_{PHO2}$ and $A_{pho2\Delta}$ values (Fig. 5A). Interestingly, some showed higher $A_{PHO2}$ than either ScPho4 or CgPho4. To identify the potential genetic basis for the divergence in Pho2-dependence, we first calculated the activity difference when one

or two regions of ScPho4 were replaced with their counterpart(s) from CgPho4. The results were plotted as two heat maps separated based on the presence of Pho2 (Fig. 5B). For example, a chimera with its NLS from CgPho4 (NLS:Cg) and the rest from ScPho4 led to estimates of the activity difference between NLS:Cg and NLS:Sc on ScPho4 background with or without Pho2.

Based on our previous results, we expected R1, NLS, and DBD in CgPho4 to enhance the activity of the chimera with and without Pho2. In particular, the first two encode CgE1 and a majority of CgE2_9aa, which we showed enhanced the activity of the main AD (Fig. 3); DBD:Cg is expected to increase the binding ability of the chimera (Fig. 2). We indeed found R1:Cg and NLS:Cg to enhance $A_{PHO2}$ on ScPho4's background (Fig. 5C, "R1, AD, NLS" group); contrary to our expectation, however, DBD:Cg reduced the chimera's activity with Pho2 (Fig. 5C, bHLH). Also unexpectedly, R1:Cg and NLS:Cg had a much smaller effect on $A_{pho2\Delta}$ (Fig. 5B bottom, 5C right), suggesting that despite their ability to increase $A_{PHO2}$, the chimeras were still dependent on Pho2. In fact, none of the 1- or 2-region swaps led to large increases in $A_{pho2\Delta}$. It is worth noting that this result also argues against a classic lock-and-key model for autoinhibition, which we expect to be broken by swapping one of the regions involved.

To quantitatively examine the contribution of R1, AD, and NLS regions from CgPho4 and whether they interact non-additively (epistasis), we fit a linear model to the data to estimate the main and interaction terms for the regions, i.e., $Y = X_0 + R1 + AD + NLS + R1:AD + R1:NLS + AD:NLS + R1:AD:NLS$. In this model, the first term represents the activity of ScPho4, the next three terms represent the main effect of each CgPho4 region (":Cg" omitted for brevity), and the rest are interaction terms. We found that R1:Cg and NLS:Cg each had significant, positive effects on their own on both $A_{PHO2}$ and $A_{pho2\Delta}$; although the magnitude was much smaller for $A_{pho2\Delta}$ (Fig. 5D, Holm-Bonferroni corrected $P < 0.05$). Both also had a significant and positive interaction term with AD:Cg on $A_{PHO2}$ (corrected $P < 0.05$ for R1:AD and AD:NLS); the estimates trended in the same direction but were not significant for $A_{pho2\Delta}$ (Fig. 5D). One explanation for the observed epistasis may be that the two CgPho4 AEDs work more efficiently with AD:Cg than with AD:Sc. However, it could also be explained by the disruption of the native conformation and function of the interacting regions resulting in a lower activity in the species-mixed constructs. Note that to minimize such effects, we designed the breakpoints to avoid any predicted secondary structures (Supplementary Figs. 1 and 2).

Since the NLS:Cg region encodes both the NLS/NES and a key part of the second AED (CgE2_9aa), we asked which of these two mechanisms was responsible for its positive effect on $A_{PHO2}$. To test the possibility that the enhanced $A_{PHO2}$ was due to a stronger nuclear localization activity of NLS:Cg, we performed fluorescence microscopy to quantify the concentration of the nuclear-localized Pho4 proteins and the ratio of nuclear vs total Pho4 proteins in six constructs that bear either NLS:Sc or NLS:Cg. No significant difference between the two groups was found (Supplementary Fig. 7, F-test $P > 0.1$). We thus conclude that the positive effect of NLS from CgPho4 on $A_{PHO2}$ was mainly due to its ability to boost activation.

We also examined the effect of swapping CgPho4's DBD. On its own, swapping DBD:Cg decreased both $A_{PHO2}$ and $A_{pho2\Delta}$ compared to ScPho4 (Fig. 5B), suggesting that DBD:Cg may be incompatible with one or multiple regions in ScPho4.

In summary, we found that R1:Cg and NLS:Cg increased the activity of the chimera with Pho2 but were insufficient to remove the Pho2 dependence alone or in combination. In the presence of Pho2, R1:Cg and NLS:Cg both showed positive epistasis with AD:Cg on ScPho4's background. Lastly, no 1- or 2-region swaps from CgPho4 into ScPho4 increased $A_{pho2\Delta}$ to the level of CgPho4. Together, we conclude that CgPho4's two AEDs can increase the activity of the chimera

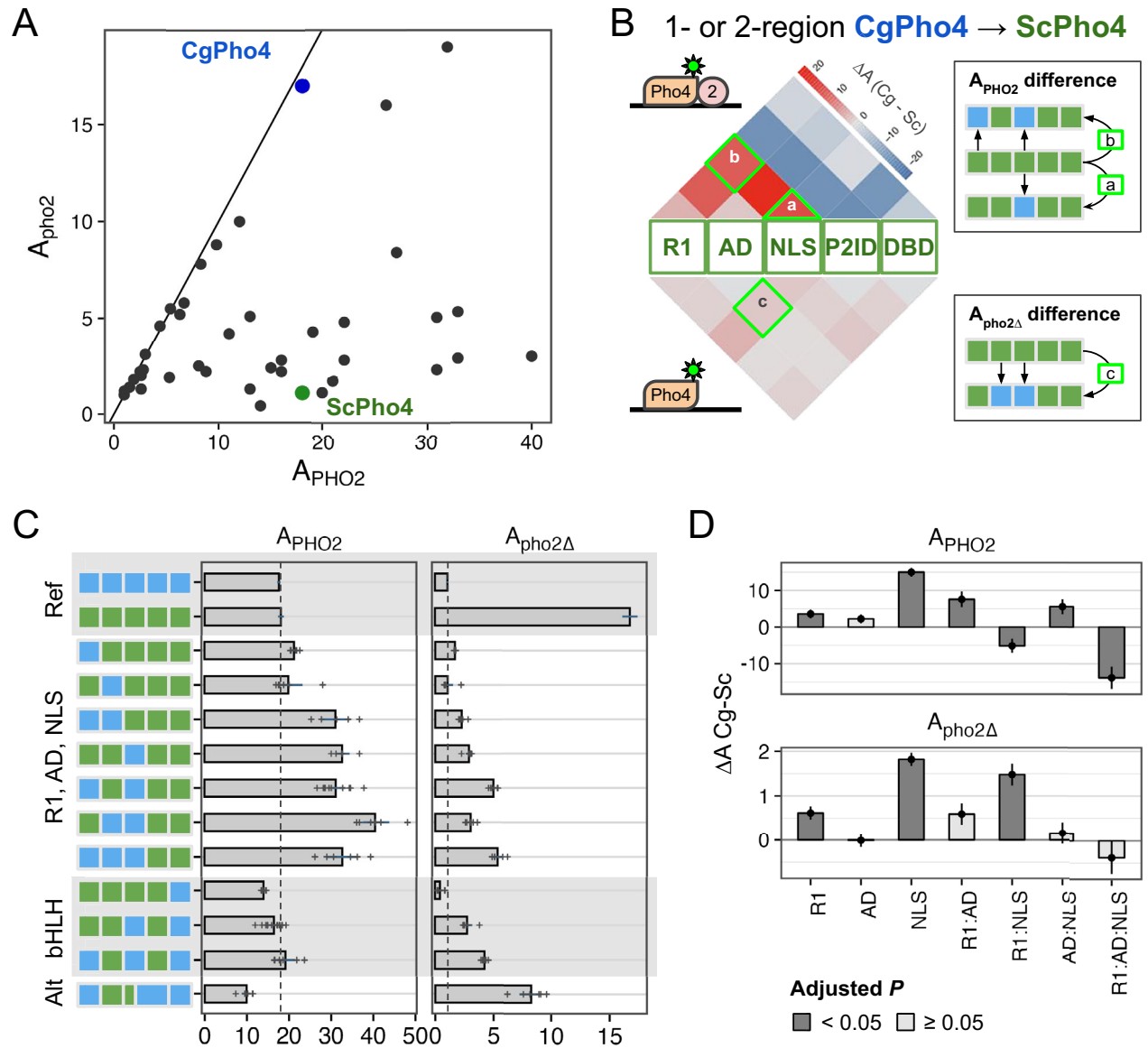

**Fig. 5 | R1 and NLS of CgPho4 confer stronger activity than their counterparts in ScPho4. A** Pho4 chimeras (black), endogenous ScPho4 (green), and CgPho4 (blue) were plotted based on their activity with and without Pho2. The solid line indicates equal activity (independent of Pho2). **B** Heatmap showing the difference in activity between the CgPho4 region(s) and the counterpart of ScPho4 measured in the ScPho4 background. The upper half shows the difference with Pho2 and the lower half without Pho2. The triangles (e.g., a) show the difference between single regions while the squares on the top show the differences when swapping two regions (see key to the right). Red color indicates a higher activity of the region from CgPho4 than the counterpart in ScPho4 and blue color indicates decreased activity. **C** A subset of the chimeras were shown with their $A_{PHO2}$ and $A_{pho2\Delta}$ values.

Bars represent the means; lines are 95% confidence intervals based on bootstrap; plus signs are individual data points (n = 6, biological replicates), which were not shown for the endogenous ScPho4 and CgPho4 (n = 36, biological replicates). Vertical dotted lines equal the endogenous ScPho4 level. **D** Epistasis analysis of CgPho4 R1, AD, and NLS. A linear model was fit to the data with all two and three-way interactions. Shown are the individual region and interaction coefficient estimates (bar and point) for the CgPho4 region(s). Line ranges centered on the estimates (bar and point) indicate the standard errors of the coefficients as calculated by the linear model fit. Dark gray indicates the term is significant at 0.05 level by student's t-test after Holm-Bonferroni correction.

in the presence of Pho2, although they fail to restore the activity without Pho2 to CgPho4's level, suggesting that additional mechanisms are at play. CgPho4's DBD, alone or in combination with R1 and the NLS region, did not contribute to increased activity.

**A double-edged sword: Pho2 interaction domain (P2ID) in ScPho4 allows for enhanced activity with Pho2 but restricts its activity without Pho2**

In the 1- and 2- region swap heatmap (Fig. 5B), the P2ID showed a puzzling pattern: swapping CgPho4's P2ID (P2ID:Cg) into the ScPho4

background had a dominant negative effect on $A_{PHO2}$ while offering little increase in $A_{pho2\Delta}$. This suggests that P2ID:Sc is essential for ScPho4's function. Conversely, swapping ScPho4's P2ID (P2ID:Sc) into CgPho4 increased $A_{PHO2}$ beyond that of CgPho4 (18 to 22) but reduced $A_{pho2\Delta}$ (17 to 4.8). This suggests that P2ID:Sc is a key factor to Pho2-dependence. To investigate the unique property of the P2ID further, we plotted all chimeras based on their $A_{PHO2}$ and $A_{pho2\Delta}$ values (Fig. 6A). Strikingly, the chimeras fell into three distinct groups based on the identity of their P2IDs. The first group had P2ID from CgPho4, and their activities were not dependent on Pho2, falling along the

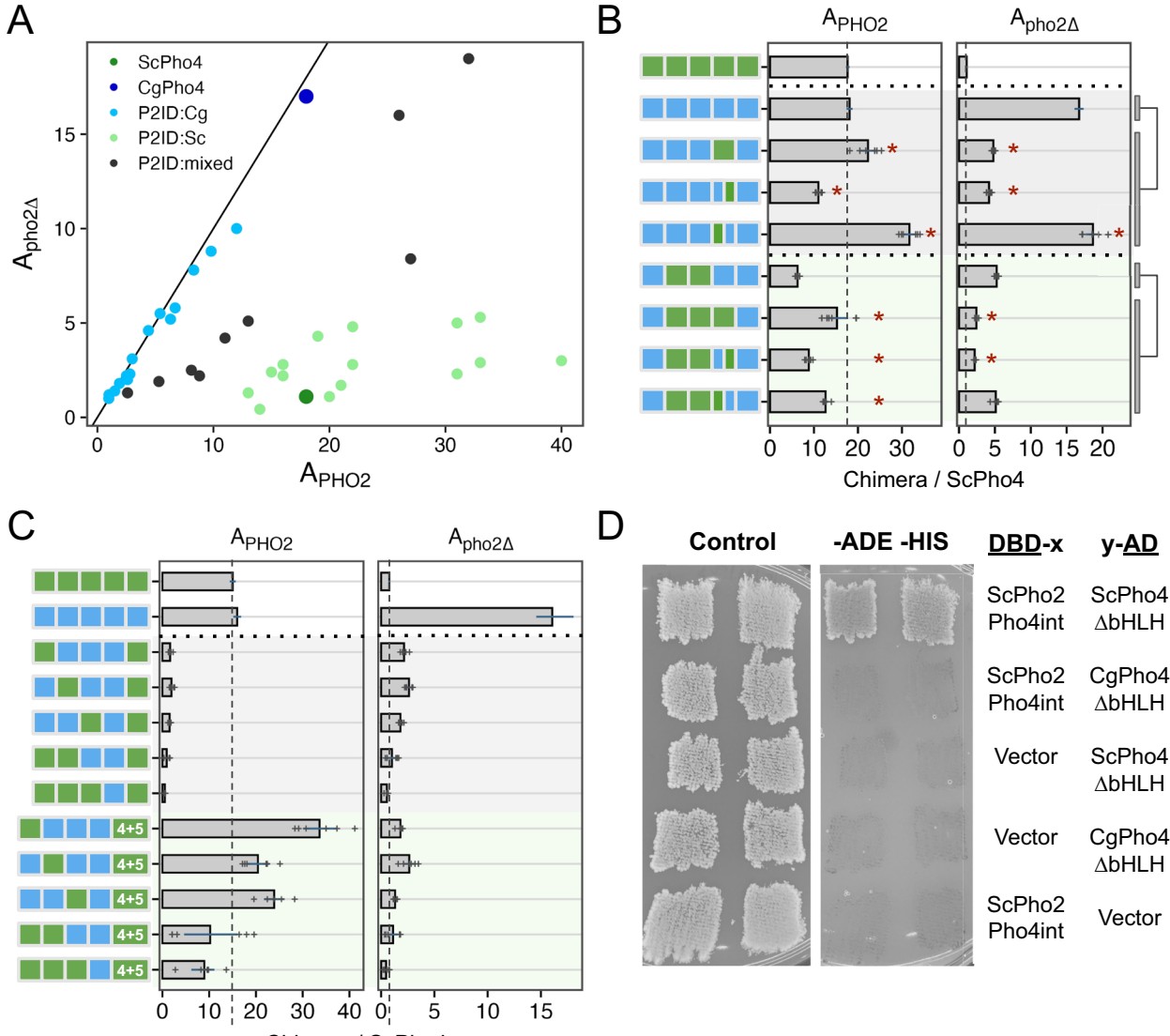

**Fig. 6 | P2ID in ScPho4 allows for enhanced activation with Pho2 via physical interaction with the co-TF. A** Scatter plot of all chimeric Pho4 constructs based on their $A_{PHO2}$ and $A_{pho2\Delta}$ values, colored by the identity of their P2ID region. Mixed P2ID uses an additional breakpoint that separates the region into two halves (Supplementary Fig. 2). **B** Comparing the effects of swapping the whole P2ID from ScPho4 into CgPho4 vs individual halves. Bars represent the means; lines are 95% confidence intervals based on bootstrap; plus signs are individual data points (n = 6, biological replicates), which were not shown for the endogenous ScPho4 and CgPho4 (n = 36, biological replicates). Two sets were tested (rows 2–5 and 6–9). Two-sided Student's t-tests were performed as indicated by the bars and lines to the right. Holm-Bonferroni correction was applied to the raw *P*-values for both sets, separately for $A_{PHO2}$ and $A_{pho2\Delta}$. An asterisk indicates corrected *P* < 0.05. **C** Testing

the hypothesis that ScPho4 DBD requires the P2ID:Sc and Pho2 to be fully functional. Plot designs, including bars, line ranges and replicate numbers are the same as in (**B**). Compared with the chimeras with P2ID:Cg + P2ID:Sc, which are all non-functional even with Pho2, adding back P2ID:Sc (4 + 5) rescued the activity with Pho2, but not without. **D** Pho4-Pho2 physical interaction was tested with a yeast two-hybrid assay. The bait and target regions were fused to the activation domain (AD) and DNA binding domain (DBD) of the yeast Gal4 protein. Interaction between them reconstitutes a functional Gal4 and allows cells to grow on the dropout media (right column), while all strains are able to grow on the rich media (left column). Vector indicates a Gal4 DBD or Gal4 AD only plasmid without a bait or target fused to it and is included as a control. Two independent transformants are shown.

diagonal line. The second group had P2ID from ScPho4. These had strong activity with Pho2 and low to no activity without Pho2, like ScPho4 does. The third group included additional chimeras with mixed P2ID regions, and they filled the intermediate space between the first two groups. Notably, many chimeras in the second group showed higher $A_{PHO2}$ than ScPho4. Most of them contained the R1 and NLS regions from CgPho4, consistent with our results above showing that these two regions enhanced the activity of the Pho4 chimeras in the presence of Pho2 (Fig. 5A, B).

The evidence above suggests that P2ID:Sc has two effects: allowing for collaborative regulation with Pho2 and restricting Pho4's activity without Pho2. Can these two functions be separated? To

answer this question, we compared the mixed P2ID chimeras to those with whole P2IDs from either Pho4. We found that swapping P2ID:Sc into CgPho4 resulted in a 1.2-fold increase in $A_{PHO2}$ and a 71% reduction in $A_{pho2\Delta}$ (Fig. 6B, row 2 vs 3). Interestingly, swapping just the second half of P2ID:Sc (aa 205–242) into CgPho4 resulted in a ~40% reduction in $A_{PHO2}$ and 75% reduction in $A_{pho2\Delta}$. By contrast, just swapping the first half of P2ID:Sc (aa 177–204) resulted in 1.76-fold increase in $A_{PHO2}$ and maintained the same level of $A_{pho2\Delta}$ (1.1-fold). From this, we deduced that the first half of P2ID:Sc mainly functions to interact with Pho2 while the second half appears to limit Pho4's activity without Pho2. The same trend was observed in another set of chimeras with lower activities (Fig. 6B, rows 6–9).

Several chimeras had very low activities even with Pho2 ($A_{PHO2} < 3.6$, or 20% of $A_{PHO2}$ for ScPho4). Most of these nonfunctional chimeras have P2ID:Cg and DBD:Sc (Fig. 6C, rows 3–7). We hypothesize that DBD:Sc requires P2ID:Sc and Pho2 to function in the context of the full length Pho4. This is contrary to the conventional view that DBDs can function on their own, which is supported by our own in vitro results (Fig. 2). Nonetheless, we reasoned that if the above hypothesis is correct, putting P2ID:Sc back by inserting it in between P2ID:Cg and DBD:Sc should rescue the non-functional chimeras in the presence of Pho2. That is what we observed (Fig. 6C, rows 8–12), with some of the chimeras even exceeding the $A_{PHO2}$ of ScPho4 and CgPho4. However, all of them still required Pho2. These results support the above hypothesis, showing that the dual-functional P2ID is essential for ScPho4 to function.

Given that chimeras with P2ID:Cg have equal activities with and without Pho2 (Fig. 6A), we asked if CgPho4 still physically interacts with Pho2 in *S. cerevisiae*. Using the yeast two-hybrid assay, we were unable to detect an interaction between CgPho4ΔDBD (aa 2–463) and a region of ScPho2 known to interact with multiple TFs, including ScPho4[32] (Fig. 6D). By contrast, ScPho4ΔDBD (aa 2–250) was able to interact with the same region of ScPho2 as previously found (Fig. 6D). This is consistent with our observation above, where chimeras with P2ID:Cg and lacking the region from ScPho4 mediating Pho2-interaction cannot be enhanced by Pho2 (Fig. 6A).

In summary, our chimera dissection revealed three key regions behind the difference in co-TF dependence. Notably, all three are IDRs. Among them, a region adjacent to the DBD in ScPho4 (P2ID:Sc) functions as a double-edged sword: it both allows Pho4 to gain activity with Pho2's help and restricts it when Pho2 is absent. We showed that these two functions are encoded by physically separate parts, offering a path for determining their respective mechanisms of actions. CgPho4 lacks the ability to interact with and use Pho2's help. Instead, the two AEDs we identified earlier—both in IDRs—conferred higher activity independent of Pho2. This, combined with the lack of the autoinhibition by P2ID:Sc, makes CgPho4 as active as ScPho4 and not dependent on the co-TF.

## Discussion

Combinatorial control plays crucial roles in eukaryotic gene regulation. Mutations disrupting TF interactions can cause dysregulation and disease[10]. However, how mutations can alter TF interdependence itself, whether in disease or evolution, is less understood. In this study, we investigated the molecular basis for natural variation in co-TF dependence in the yeast phosphate starvation (PHO) response. We found three key differences between two orthologous Pho4s with varying dependence on the co-TF Pho2 (Fig. 7): (1) <u>DNA Binding Affinity</u>: CgPho4 binds the same consensus DNA motif with 3–4-fold higher affinity than ScPho4; (2) <u>Activation potential</u>: CgPho4 has two unique activation enhancing domains (AEDs) that increase the activation potential of both Pho4s. (3) <u>Autoinhibition</u>: ScPho4 contains an IDR next to its DBD that both allows it to interact with Pho2 to gain enhanced activity, and inhibits its activity in the absence of Pho2. Therefore, our results support both the enhanced activity model and the autoinhibition model for the difference in Pho2-dependence between Pho4 orthologs.

Among the three differences, the contribution of CgPho4's two AEDs to the TF's activity and dependence on Pho2 is well supported by the yeast one-hybrid data and chimeric Pho4 results (Figs. 3, 5). We are not aware of existing examples or proposed mechanisms for such a phenomenon. One hypothesis for how AEDs work is that they are weaker ADs with the same biochemical activities, i.e., recruiting cofactors through protein-protein interactions. As such, they are not sufficient for activation by themselves but can increase the activity of the nearby AD (the effect can be non-additive). Alternatively, AEDs may affect the conformation of the AD and have no activity on their own. Further tests by synthetic constructs and biochemical assays, such as

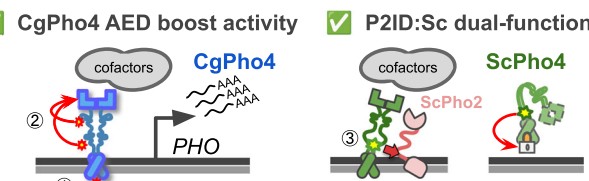

**Fig. 7 | Summary of Pho4 protein divergence and its contribution to the Pho2-dependence variation in ScPho4 and CgPho4.** Left (1): CgPho4 DBD recognizes the same DNA sequence as ScPho4 DBD, but binds the consensus motif with a 3–4 fold higher affinity. While consistent with the enhanced activity model, further studies found no evidence for the enhanced activity to be a major factor for the reduced Pho2-dependence. (2) Two regions in the IDR of CgPho4 (red asterisks) can enhance the activity of the main activation domain (AD) but are not sufficient on their own to activate gene expression. These two Activation Enhancing Domains (AEDs) directly contribute to CgPho4's reduced Pho2-dependence. Right (3): An IDR (green asterisk) adjacent to the DBD in ScPho4, named P2ID, can both interact with ScPho2 and boost the activity of Pho4 and also repress its activity when Pho2 is absent, providing support for the autoinhibition model.

co-IP followed by mass-spec, will provide mechanistic insight into how AEDs function.

By contrast, the significance of the binding affinity differences between the two Pho4s remains unclear (Fig. 2E). Binding kinetics predicts that if the nuclear concentrations of both Pho4 are much higher than their $K_D$, a 3–4-fold difference in affinity would have little impact. Conversely, if the nuclear concentration of Pho4 is close to its $K_D$, the same difference could significantly affect gene induction. Existence of the second scenario is supported by a study showing that nuclear ScPho4 levels are much lower at intermediate phosphate concentrations than in extreme starvation conditions, leading to differential binding and induction of ScPho4's targets[33]. In our chimeric Pho4 experiments, replacing ScPho4's DBD with that from CgPho4 reduced Pho4's activity rather than increasing it both with and without Pho2 (Fig. 5). It seems to suggest that the binding affinity difference has no functional impact in vivo. However, it is worth noting that we assayed the activity of chimeric Pho4s in the *pho80Δ* background where all Pho4 proteins are constitutively inside the nucleus at near-maximal levels, hence not allowing us to test the effect of the affinity difference at a lower level of nuclear Pho4. Future studies will need to measure Pho4 activities at varying nuclear concentrations to test the above hypothesis.

Surprisingly, we found that ScPho4's DBD requires both ScPho4's P2ID and Pho2 for full activity (Fig. 6C), challenging the view that TFs are modular, where DBD are expected to function independently. Although ours and others' studies confirmed that ScPho4 DBD can bind DNA in vitro (Fig. 2) and even in vivo when overexpressed on its own[34], they may not fully reflect its physiological activity, which depends on the nuclear concentration, nucleosomal context, and interactions with other TF regions and cofactors. Further experiments are needed to resolve the paradox and re-examine the assumption of TF modularity.

Only one of the three functional differences identified between the two Pho4 orthologs is in a structured region. The two AEDs and the dual-functional P2ID:Sc are both within IDRs. IDRs are abundant in eukaryotic TFs, with over 80% having at least one such region[35]. Approximately 75% of ScPho4 and CgPho4 were predicted to be IDR (Supplementary Fig. 1). While most studies of TF function and evolution focus on the structured regions like the DBD, our work joins a small number of studies highlighting the significance of IDR divergence in TF evolution[36,37]. On one hand, IDRs play crucial roles in nearly all aspects of a TF's function, including recruiting cofactors, interacting with co-TFs, contributing to binding specificity and forming molecular condensates[38–43]. On the other hand, IDRs evolve much faster than

structured regions, potentially offering more raw variation for natural selection to act on. Our current understanding of how IDR evolves and affects TF function is limited by challenges in aligning their sequence and studying their functions. Progress in experimental techniques like phase separation[43] and large language models trained on protein sequences offers new opportunities to investigate TF IDR's function and evolution[44].

Our study focused on dissecting the genetic basis for Pho2-dependence variation in two Pho4 orthologs. We previously showed that this trait varies across Pho4 orthologs, with reduced Pho2-dependence potentially evolving independently in more than one lineage[15]. We wondered whether the IDR-associated divergence identified above also correlated with the level of Pho2-dependence more broadly. Preliminary analyses of the activation potential and P2ID length across eight Pho4 orthologs with different Pho2-dependence levels supported this speculation (Supplementary Fig. 8).

Lastly, our study illustrates a mechanism for the evolution of combinatorial control in eukaryotes. Unlike previous research that focused on the gain and loss of TF interactions in gene regulatory networks[45,46], we show that the TF's dependence on the co-TF itself can evolve. This divergence can lead to significant rewiring of the regulatory network, as seen with Pho4, where reduced co-TF dependence correlates with an expanded target gene network[15]. This contrasts with the dominant pattern seen in the literature on combinatorial control evolution: the Johnson lab, for example, showed that changes in TF combinations in the yeast mating type pathway altered the mode of regulation but maintained the overall network output[11,12,47,48]. While cases of TF interdependence evolution are still rare, it is interesting to compare our example with Gain-of-Function mutations in kinases: in the proto-oncogene ABL, a fusion with another gene called BCL makes the merged protein independent of the upstream and downstream activators, which drives excessive cell proliferation and leads to Chronic Myeloid Leukemia[49]. Whether similar changes in TF dependence on co-TFs lead to misregulation and in turn causes diseases or novel phenotypes is an interesting question for future studies.

In summary, our study provides a detailed molecular picture of how co-TF dependence is mediated and how it evolves, particularly through IDR changes. Further exploration of both questions is essential for understanding gene regulation and regulatory evolution in eukaryotes.

## Methods
Breakpoints used for the chimeric Pho4 constructs and individual regions are listed in Table 1. Plasmids and strains are listed in Supplementary Data 1 and Table 2. Computational and statistical analysis scripts performed in this study are available at https://github.com/binhe-lab/E013-Pho4-evolution, which will be archived using Zenodo and minted with a DOI at the time of publication.

### Bioinformatic analyses of Pho4 orthologs
Pho4 ortholog sequences were from the Yeast Gene Order Browser (http://ygob.ucd.ie/), and were aligned using ProbCons[50] via JalView's Web Service[51,52]. The alignment was manually edited to align the five Pho80/85 motifs[24]. Secondary structures for ScPho4 and CgPho4 were predicted using the PSIPRED 4.0[53]. 9aaTAD motifs were predicted using the webapp https://www.med.muni.cz/9aaTAD/[31]. For ScPho4, one match was found using the moderately stringent pattern, located at aa 75–83; for CgPho4, one match was identified using the most stringent pattern, at aa 270–278, with three more using the moderately stringent patterns (aa 20–28, 24–32, 282–290). Among the latter three, aa 24–32 had a lower % match at 67% vs 83% for the others.

### Identifying Pho4 binding motifs from ChIP-seq data
Chromatin Immunoprecipitation (ChIP-seq) for ScPho4 and CgPho4 were previously performed in *S. cerevisiae*, with both Pho4 expressed from the same endogenous ScPho4 locus with native regulatory sequences[15]. ChIP identified peaks for both Pho4s were downloaded from the supplementary files of the above publication. The sequences under each peak were extracted from *S. cerevisiae* genome (sacCer3, NCBI refseq assembly GCF_000146045.2), which were submitted to the peak-motifs tool without control sequences on the RSAT Fungi server (https://rsat.france-bioinformatique.fr/fungi/)[54]. The top motif was reported for each Pho4 ortholog.

### Protein expression and purification for DNA binding assays
All recombinant proteins were expressed using pET-11a (Sigma #69436-3) based vectors in BL21(DE3) *E. coli* cells. Vector maps are available upon request. The DNA binding domain (DBD) constructs included ScPho4 DBD-6xHIS (aa 236–312) and CgPho4 DBD-6xHIS (aa 452–533) cloned downstream of the T7 promoter in pET-11a. The transformed bacterial strains were grown overnight in LB + 100 ug/mL ampicillin. The overnight culture was diluted to OD 0.1 and the cells were grown to OD 0.6 and induced with 1 mM IPTG for 2 h. Induced cells were collected by centrifugation at 6000 rpm at 4 °C for 35 min, and were snap frozen in liquid nitrogen, then stored at −80 °C until purification. Our preliminary experiments showed that the protein of interest was largely in the insoluble fraction. Therefore, we performed a refold protocol on the Ni-NTA column. Briefly, frozen pellets were resuspended in 1xPBS, pH7.4 with cOmplete EDTA-free protease inhibitor (Sigma #11836170001) and sonicated for 45 cycles of 1 sec on / 2 sec off repeated 3 times at 50% power to lyse the cells. Sonicated samples were centrifuged at 35k rpm for 30 min. The supernatant was discarded, and the pellet was resuspended in the solubilization buffer (20 mM TrisBase, 0.5 M NaCl, 5 mM imidazole, 5.5 M Guanidine Hydrochloride, 1 mM 2-mercaptoethanol pH 8.0) and stirred at -25 °C for an hour. The solubilized pellet was centrifuged at 35k rpm for 30 min, and the supernatant was filtered and loaded onto a 5 mL Ni-NTA column. The column was washed with a urea buffer (20 mM TrisBase, 0.5 M NaCl, 20 mM imidazole, 5.5 M Urea, 1 mM 2-mercaptoethanol pH 8.0). Then, the protein was refolded with a reverse urea gradient from buffer A (20 mM TrisBase, 0.5 M NaCl, 20 mM imidazole, 5.5 M Urea, 1 mM 2-mercaptoethanol pH 8.0) to buffer B (20 mM TrisBase, 0.5 M NaCl, 20 mM imidazole, 1 mM 2-mercaptoethanol pH 8.0) on a

## Table 2 | Yeast strains

| ID | Genotype | Source |
|---|---|---|
| yH373 | pho4::NAT; pho2::HIS3; pho80::TRP1; pho5::mCherry | this study |
| yH555 | pho4::NAT; his3::pRS303-HIS3; pho80::TRP1; pho5::mCherry | this study |
| yH549 | gal4delta gal80delta LYS2::GAL1-HIS3 GAL2-ADE2 MET2::GAL7-lacZ | PMID: 8978031, gift from Fassler lab |
| yH696 | gal4delta gal80delta LYS2::GAL1-HIS3; GAL2-ADE2; MET2::GAL7-lacZ; ura3::pRS306-*GAL1pr*-mCherry-URA3 | this study |
| yH156 | pho80::TRP1 | PMID: 28485712 |
| various | chimeric Pho4 plasmids transformed into yH373 and yH555 | this study |
| various | yeast two-hybrid plasmids transformed into yH696 | this study |
| various | yeast one-hybrid plasmids transformed into yH696 | this study |

BioRad FPLC. The refolded protein was eluted off of the Ni-NTA column using a gradient from buffer C (20 mM TrisBase, 0.5 M NaCl, 20 mM imidazole, 1 mM 2-mercaptoethanol pH 8.0) to buffer D (20 mM TrisBase, 0.5 M NaCl, 500 mM imidazole, 1 mM 2-mercaptoethanol pH 8.0). Fractions containing the protein of interest were identified by gel electrophoresis, pooled, and diluted with a no salt buffer (25 mM $Na_2HPO_4$ pH 7.0, 0.5 mM THP) and run on a Heparin column equilibrated with low salt buffer (25 mM $Na_2HPO_4$ pH 7.0, 0.15 M NaCl, 0.5 mM THP). Protein was eluted with a gradient from low to high salt buffer (25 mM $Na_2HPO_4$ pH 7.0, 1.5 M NaCl, 0.5 mM THP). Protein-containing fractions were concentrated using a 3 kDa cutoff Amicon centrifugal filter (Sigma UFC8003) and loaded onto a Superdex 75 size exclusion column equilibrated with a storage buffer (25 mM $Na_2HPO_4$ pH 7.0, 0.5 M NaCl, 0.5 mM THP). Fractions containing the expected size products were collected, analyzed by gel electrophoresis, and stored in the storage buffer at 4 °C.

N-GST-CgPho4 full length was constructed for the Protein Binding Microarray assay. The corresponding N-GST-ScPho4 purification has been described before[55]. Both proteins were grown and lysed as described in ref. 55. Briefly, BL21 *E. coli* cells containing the constructs were induced at OD600 0.8 with 1 mM IPTG and collected by centrifugation after 3 h of induction at 30 °C. Pellets were snap frozen in liquid nitrogen and stored at −80 °C until purification. Cells were lysed with rLysozyme (Millipore 71110) for 20 min at room temp in the lysis buffer (1xPBS, pH7.4) with the cOmplete protease inhibitor tablet and 1 mM PMSF. The protein was run on a GST column equilibrated with the lysis buffer and eluted with a gradient to buffer B (50 mM Tris, 10 mM glutathione, pH 8). GST-CgPho4 fractions containing the protein were pooled and loaded onto a heparin column equilibrated with low salt buffer (25 mM $Na_2HPO_4$ pH 7.4, 150 mM NaCl, 0.5 mM THP) and eluted with a gradient to high salt buffer (25 mM sodium phosphate dibasic pH 7.4, 1.5 M NaCl, 0.5 mM THP). Fractions containing the protein were pooled and concentrated with a 30 kDa cutoff Amicon centrifugal filter (Sigma UFC9030) and loaded onto a Superdex 200 size exclusion column equilibrated with the storage buffer (25 mM HEPES pH 7.4, 500 mM NaCl, 0.5 mM THP). Protein was run on an SDS-PAGE and pure fractions were pooled and concentrated. 10% glycerol was added before snap freezing and storage at −80 °C.

### Universal protein binding microarray (uPBM)

The uPBM was performed following the PBM protocol as described in ref. 56. Briefly, after the primer extension step is used to double-strand the DNA molecules on the array, the chambers are blocked with 2% milk. After washing, proteins are incubated with the array for 1 h. Alexa Fluor 488-conjugated anti-GST antibody (Invitrogen A-11131) was used to detect binding. The array was scanned using a GenePix 4400 A scanner (Molecular Devices). GST-ScPho4 and GST-CgPho4 (full length) were prepared as described above. Both proteins were assayed at a final concentration of 1 µM as determined by the optical absorbance. An 8 × 15k array was used to assay all possible 9-mers, from which a robust 7-mer enrichment score is derived. The non-parametric enrichment score, or E-score, is invariant to differences in the concentration of the proteins used in the assay, and thus are suitable for comparisons of relative affinities between arrays. E-score ranges from −0.5 (lowest enrichment) to +0.5 (highest enrichment). Scores greater than 0.35 correspond to specific TF-DNA binding[29,57]. Data analysis was performed using custom Perl scripts as described in ref. 56 to extract and normalize fluorescence based intensity.

### Genomic context protein binding microarray (gcPBM)

All probes in the DNA library are 60 bp in length with 24 bp complementary to the primer used for double stranding and 36 bp of genomic region centered on the E-box motif or its variant. The library contains (1) 5711 *S. cerevisiae* genomic regions with putative Pho4 binding sites; (2) 150 negative controls from the *S. cerevisiae* genome not specifically bound by Pho4[55]; (3) 4000 DNA sequences used in previous MITOMI experiments to calibrate the binding affinities[58]; (4) 150 genomic regions from *C. glabrata* that contain Pho4 consensus and nonconsensus binding sites[15]; (5) 100 probes from the library of sequences we tested with Biolayer Interferometry; and (6) 150 negative controls from the *C. glabrata* genome not bound by CgPho4. We used the NNNNGTG, CACNNN, and GTGNNN libraries from Maerkl and Quake (2007) in our gcPBM design. The MITOMI and BLI probes required the addition of random flanks to maintain the 36 bp length;10 different flanking sequences were generated for each sequence. Each probe is represented by six replicates, including three replicates in each orientation and were distributed randomly across the array. The custom 8 × 60k (8 chambers, 60,000 DNA spots per chamber) was synthesized by Agilent Technologies. The gcPBM was performed and analyzed following the PBM protocol as described above and in refs. 29,55. The log transformed median intensity of the 6 replicate probes were used for comparisons. Because the log signal intensities are not directly comparable between the two Pho4 proteins, which were assayed on separate arrays, we used Spearman's rank correlation coefficient to quantify the level of concordance in their sequence preference. Any difference in preference should result in a change in the ranks within each Pho4's data.

### Biolayer interferometry (BLI)

A library of 17-bp dsDNA was used for the assay. The consensus probe had the sequence "CTAGTCC**CACGTG**TGAG", with the E-box motif bolded, and was identical to the DNA used in the crystal structure of ScPho4's DBD[26]. Nine half-site variants, including "AACGTG, TACGTG, … CAAGTG, ACGTG, CATGTG" were constructed on this background. For each probe, the complementary ssDNA oligos were synthesized by Integrated DNA Technologies (IDT). One of the two probes was biotinylated on the 5' end. To anneal them into dsDNA, 1 pmol (5 µL of 200 µM) of the biotinylated oligo was mixed with 2 pmol (10 µL of 200 µM) of the complementary, unmodified strand in the Nuclease-Free Duplex Buffer (30 mM HEPES, pH 7.5; 100 mM potassium acetate). The mixture was heated to 95 °C and then left at room temperature for it to cool down. Annealed probes were stored at −20 °C until use.

ScPho4 DBD and CgPho4 DBD were purified and stored as described above. Protein quality was checked weekly for signs of degradation using Dynamic Light Scattering (DLS) on a DynaPro NanoStar instrument. Before each experiment, the concentration of the protein prep was measured in triplicates using a NanoDrop instrument. The mean concentration was used to prepare the protein dilutions and calculate $K_D$.

BLI experiments were performed on an Octet RED 96 instrument at 30 °C with 1000 rpm shaking. For each protein-probe pair, eight streptavidin (SA) biosensors were hydrated for 15–20 min at room temperature in the 1× kinetics buffer (1× PBS pH 7.4, 0.01% BSA, 0.002% Tween-20, 0.005% Sodium azide). A black 96 well flat-bottom plate was loaded with experimental components also diluted in the 1× kinetics buffer. To begin the experiment, biosensors were equilibrated in a 1× kinetics buffer for 60 s to reach the baseline. Seven biosensors were then submerged in 35 nM biotinylated annealed DNA for 30 s, while one biosensor was submerged in the 1× kinetics buffer as a no-DNA control. Biosensors were then submerged in 1 µg/ml biocytin for 60 s to block any empty streptavidin pocket on their surface, before being dipped back into a 1× kinetics buffer for 60 s for a baseline measurement. Loaded and blocked biosensors were then submerged into a gradient of protein concentrations calculated for each probe based on the $K_D$. To obtain reliable measurements, we use a concentration range that spans 10× to 1/10 of the $K_D$ values. Biosensors stayed in the protein solution for 900–1000 s or until equilibrium was reached.

Data analysis was performed in the ForteBio Data analysis v11 software. After subtracting the background and aligning the y-axis, the processed data were subjected to either a steady state analysis or a kinetic curve fitting. For steady state analysis, the equilibrium-level signal from each biosensor was plotted against the protein concentration, from which $K_D$ was calculated. Kinetic curve fitting was done using a one-site specific binding with Hill slope model as implemented in the Data analysis v11 software. The latter more effectively fit the BLI data and thus were utilized for most of the analysis. For the consensus sequence, we found the kinetic curve-based estimates for both ScPho4 and CgPho4 to have higher variance, likely due to the fast kinetics not adequately captured by the model[59]. Steady-state analysis gave consistent mean $K_D$ as the kinetic analysis did, but resulted in lower variance, and was used instead. Two to four replicates were performed for each 17 bp DNA library sequence, using at least two independent protein preps.

### Electrophoretic mobility shift assay (EMSA)

IR700 labeled 17 bp consensus DNA (same as the consensus DNA for BLI above, Integrated DNA technologies) was diluted to 0.1 nM and mixed 1:1 with a 2× dilution series of the DBD of interest starting at 32 nM in binding buffer (20 mM Tris-HCL, pH 8, 150 mM KCl, 10% glycerol, 5 mM MgCl2, 1 mM EDTA, and 1 mM DTT). The mixture was incubated for 1 h at 4 °C. A 10% native PAGE gel in 1× Tris Glycine buffer (2.5 mM Tris pH 8.3, 19.2 mM glycine) was prerun at 200 V for 20 min. The DNA/protein mixture was then loaded onto the gel and ran for 25–35 min. The gel was imaged using the Odyssey FC imager in the 700 nm channel. To estimate $K_D$, the unbound band in each protein-containing lane was quantified using the ImageStudio software (with background subtraction) and divided by the no protein control. This value was then subtracted from 1 and plotted against the protein concentrations. A nonlinear curve fitting was performed in Prism v10.2.1 using the one site specific binding model. $K_D$ estimates were reported.

### Yeast media and growth

Yeast cells were grown in Yeast extract-Peptone Dextrose (YPD) medium or Synthetic Complete (SC) medium, using Yeast Nitrogen Base without amino acids (Sigma Y0626) supplemented with 2% glucose and amino acid mix. Phosphate starvation medium was made using Yeast Nitrogen Base with ammonium sulfate, without phosphates, without sodium chloride (MP Biomedicals, 114027812) and supplemented to a final concentration of 2% glucose, 1.5 mg/ml potassium chloride, 0.1 mg/ml sodium chloride, and amino acids, as described previously[15]. Phosphate concentration in the medium was measured using a Malachite Green Phosphate Assay kit (Sigma, MAK307).

### Yeast strain and plasmid construction

The hosts containing the endogenous *PHO5*pr-mCherry reporter were constructed by replacing the endogenous coding sequence with mCherry using CRISPR/Cas9. Briefly, guide RNAs targeting *PHO5* were designed in Benchling (https://www.benchling.com) and cloned into bRA89 (AddGene 100950), a plasmid that contains the Cas9 protein and gRNA scaffold[60]. Homology arms to the *PHO5* 5'UTR and 3' UTR were added onto an mCherry donor DNA using PCR. The donor DNA and plasmid were co-transformed into the host using the standard LiAc transformation protocol. Transformants were selected for the CRISPR plasmid using hygromycin resistance and screened with PCR. Positive clones were validated using Sanger sequencing and fluorescence microscopy. Pho2 was then knocked out using a *HIS3* cassette with homology arms to the *PHO2* 5'UTR and 3'UTR.

The chimeric Pho4 plasmid library was constructed using Golden Gate. The fragments of CgPho4 and ScPho4 were PCR amplified using

Phusion Flash polymerase (Thermo Scientific F548S) with unique overhangs for Golden Gate. They were then assembled and inserted into a pRS315-based backbone that contains the Sc*PHO4* promoter, a C-terminal in-frame mNeon tag followed by the Sc*PHO4* 3' UTR and terminator. Primers were designed using the NEBridge Golden Gate assembly tool. The inserts were verified with PCR and Sanger sequencing and then transformed into the yeast hosts using either the standard LiAc protocol[61] or the Zymo yeast transformation kit (Zymo research T2001) and plated onto SD-leu media.

### Flow cytometry

Cells were inoculated into a 96 deep-well plate (Fisher Scientific 07-200-700) and grown overnight in SD-Leu or SC medium supplemented to final concentrations of 0.13 mg/ml adenine and 0.1 mg/ml tryptophan to reduce autofluorescence[62]. Cultures were diluted in the morning to an OD600 of 0.15 and grown to an OD600 of 0.6. These cells were directly subjected to flow cytometry at a flow rate of 25 uL/min on an Attune NxT flow cytometer (ThermoFisher) fitted with an autosampler. Data were collected using the Attune NxT software v3.1 for FSC, SSC and the appropriate fluorescence channels. Calibration beads (Spherotech RCP305A) were run routinely to ensure that experiments from different days were comparable. Pho4-mNeon was measured in the BL1 channel with 488 nm excitation and $510 \pm 10$ nm emission; mCherry was measured in the YL2 channel using 561 nm excitation and $615 \pm 25$ nm emission. Voltages for each channel were set by the brightest sample and negative control so the sample signals were between $10^2$–$10^5$. Events were gated based on FSC-H / SSC-H to remove non-cells, then on the FSC-W / FSC-H to isolate singlets (Supplementary Fig. 9). At least 10,000 gated events were collected per sample. Each strain was measured at least three times, and on two different dates. A nonfluorescent strain was included in every experiment for subtracting the autofluorescence. Further gating and analyses were performed in R using the FlowClust and FlowCore packages. Detailed analysis scripts are available in the project github repository.

### Fluorescent microscopy for Pho4 nuclear localization

Six Pho4 constructs were chosen, with three bearing NLS:Sc (SSSSS, CCSCC, SSSCC) and three bearing NLS:Cg (SSCSS, CCCCC, SSCCS). They were transformed into either a *PHO2* wild type or *pho2Δ* background. Yeast cells were grown to mid-log phase in SD-Leu and fixed using 4% paraformaldehyde for 10 min, washed twice with 1xPBS before DAPI staining. 4',6-diamidino-2-phenylindole (DAPI, Sigma, D9542) was diluted in the respective medium and added to the fixed cells at a final concentration of 10 µg/mL. Cells were incubated with DAPI in the dark for 30 min before DAPI was removed and cells were washed twice with 1xPBS. Fixed cells were mounted on a 1.2% agarose pad on a depression slide for image. Fluorescent imaging was performed on a Leica epifluorescence microscope. A 405 nm laser was used for DAPI excitation and 430–550 nm for emission. For Pho4-mNeON, 488 nm was used for excitation and 530 nm for emission. A bright field image was also taken to show the cell boundaries. A total of three images were recorded for each field of view.

Microscopy analysis was performed in ImageJ (v 1.53). To quantify the nuclear fraction of Pho4 proteins, ten cells were randomly selected for each construct in each host background, and the experiment was repeated two times, resulting in a total of 15 or 20 cells quantified in *PHO2* and *pho2Δ* backgrounds, respectively. Cell boundaries were manually traced using the bright field channel, and nucleus using the DAPI channel. Both were added to the ROI manager, and the raw integrated density (sum of pixel values) was quantified for both areas. The ratio of nuclear vs whole cell integrated density was used for plotting and statistical analysis.

## Yeast one-hybrid and yeast two-hybrid

Indicated regions from CgPho4, ScPho4, and ScPho2 were PCR amplified and cloned into pGBD-C3[63] containing the Gal4 DBD using Gibson Assembly. Plasmids were verified with PCR and Sanger sequencing. The *GAL1pr*-mCherry reporter was created using a pRS306 based integrative plasmid digested with StuI and inserted into the *ura3* locus of a *gal4Δ gal80Δ S. cerevisiae* strain. The reporter was tested using flow cytometry according to the protocol above using only the red (YL2) channel. Plasmids were then transformed into this yeast strain using the standard LiAc method[61] or the Zymo yeast transformation kit (Zymo research T2001) and selected for on SD-trp. For the yeast two-hybrid, the Gal4 activation domain (AD) and Gal4 DNA binding domain (DBD) fusion plasmids were constructed by PCR amplifying the indicated regions and cloning into pGBD-C3 and pGAD-C3[63] using Gibson Assembly. The plasmids were co-transformed and selected for on SD -leu -trp. Positive colonies were patched onto fresh plates and grown at 30 °C for 24 h before replica plating onto SD -leu -trp -ade -his to test for interactions. Plates were imaged after 45–50 h.

## Statistical analyses

All replicates are biological. For yeast strains, the same strain was grown and measured in separate vials. Independent transformants were tested for the flow cytometry host strains and randomly selected constructs and were found to generate consistent results. For recombinant proteins, the same constructs were independently transformed into the bacteria and separate batches of purified proteins were used for the measurements.

*Binding affinity comparison by BLI* (Fig. 2): $K_D$ estimates from either steady state or kinetic curve fitting analyses were log10 transformed and a Student's t-test was used to compare ScPho4 vs CgPho4 DBDs against the same 17-bp DNA. Raw two-sided *P*-values and Holm-Bonferroni corrected *P* values for the 10 tests were reported; *Activation* (Fig. 3): median fluorescence intensity (MFI) for the *GAL1pr*-mCherry reporter was recorded from flow cytometry for each sample. For Fig. 3C, each Gal4 DBD fusion construct was compared to the background level. Because the host (with *GAL1pr*-mCherry but no Gal4 DBD plasmid) and Gal4 DBD alone showed a similar level of low MFI, we combined them as the reference group. A linear model was fit in R with the following command "lm(MFI ~ Genotype)", where "Genotype" is a factor representing the constructs. This model estimates a common standard deviation for all constructs and tests the significance of each construct against the reference group with a two-sided t-test. The raw *P*-values were corrected for multiple testing using the Holm-Bonferroni procedure. Constructs with a corrected $P \leq 0.05$ were considered significant. For Fig. 3D, the first two groups were tested using a two-sample t-test, while the third involved multiple levels, and were tested as described above for Fig. 3C. A Holm-Bonferroni correction was applied to the raw *P*-values for all six tests together. *Epistasis between regions R1, AD, and NLS of CgPho4* (Fig. 5D): to determine the main and interaction terms in these swaps from CgPho4 on ScPho4's background, we fit a linear model, $Y = X_0 + R1 + AD + NLS + R1{:}AD + R1{:}NLS + AD{:}NLS + R1{:}AD{:}NLS$ in R, using the command "lm(A ~ R1 * AD * NLS)", where "A" is either $A_{PHO2}$ or $A_{pho2\Delta}$, and the independent variables are coded as ScPho4 = 0, CgPho4 = 1. The raw *P*-values were adjusted for multiple testing using the Holm-Bonferroni procedure. *Split P2ID* (Fig. 6B): The same procedure as applied to the activation region test above (Fig. 3) was applied to the split P2ID swaps. Two sets of four chimeras were chosen, each with a reference construct having P2ID:Cg, one with the entire P2ID swapped for the ScPho4 version, and two with the first or second half of P2ID swapped for the ScPho4 version. The latter three were compared to the reference using a linear model "lm(MFI ~ Genotype)", for $A_{PHO2}$ and $A_{pho2\Delta}$ separately. Holm-Bonferroni correction was applied to the two sets combined (6 tests in total), again separately for $A_{PHO2}$ and $A_{pho2\Delta}$. A corrected $P < 0.05$ was considered significant. *Nuclear fraction of Pho4 chimeras* (Supplementary Fig. 7): The ratio of the sum of pixel values in the GFP channel inside the nucleus vs the whole cell was treated as the response variable, and the identity of the NLS region ("Cg" vs "Sc") as the predictor. First, a two-way ANOVA was performed with the formula "lm(Nuc_frac ~ NLS + Host)", where Host is either *PHO2* or *pho2Δ*. Second, given that ratios are often non-normally distributed, we performed the non-parametric Kruskal-Wallis test with the command "kruskal.test(Nuc_frac ~ NLS)" separately in the *PHO2* and *pho2Δ* hosts. The ANOVA F-test *P*-value (0.13) was reported in the results. The two *P*-values for the second test were 0.42 (*PHO2*) and 0.36 (*pho2Δ*).

## Reporting summary

Further information on research design is available in the Nature Portfolio Reporting Summary linked to this article.

## Data availability

Source data for figures in this paper are provided as a Source Data file and are available at https://doi.org/10.5281/zenodo.14501732. Raw microscopy images for quantifying Pho4 nuclear concentration are available at https://doi.org/10.6084/m9.figshare.28437011. Protein Binding Microarray data are available through the Gene Expression Omnibus (GEO) under GSE293214 and GSE293355. No restrictions apply to any of the data generated in this study. Source data are provided with this paper.

## Code availability

Scripts for figures and statistical tests are available at https://doi.org/10.5281/zenodo.14501732.

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

## Acknowledgements

We would like to thank Dr. Miles Pufall's lab for teaching us EMSA. We thank Dr. Jan Fassler for sharing many yeast plasmids for the yeast one- and two-hybrids. Drs. Jan Fassler, Miles Pufall, Craig Ellermeier and Todd Washington all critically read the thesis chapter by L.F.S., on which this manuscript is based. We thank Kyle Malcolm for helping with the early development of the BLI assay. Christian Weinrich discovered the P2ID's effect on Pho2-dependence during his rotation. We thank Dr. Yann Vanrobaeys for helping with an initial analysis using PADDLE. We would like to acknowledge the use of resources at the Protein and Crystallography Facility within the Carver College of Medicine at the University of Iowa and thank Lokesh Gakhar, Zhen Xu, and Devin Reusch for assistance with protein purification and BLI assays. This work was primarily supported by NIH R35-GM137831 (to B.Z.H.). L.F.S. was supported on NIH Predoctoral Training Grant T32GM008629; B.J.B. was supported on NIH Predoctoral Training Grant T32GM1454441; R.G. was supported by NIH R01-GM135658 and B.Z.H. was also supported by a startup fund from the University of Iowa.

## Author contributions

L.F.S. and B.Z.H. designed the experiments. L.F.S. and E.M.O. constructed the chimeras and performed the flow cytometry. L.F.S. performed the EMSA and the yeast two hybrid experiments. J.L. performed the microscopy for nuclear localization analysis. E.M.O. and B.J.B. performed experiments and analyses for the revision. Y.Z. and W.Z. performed the PBM experiments with input from R.G. J.Z. and T.H.C. established critical methods. X.Z. generated preliminary results for the work. N.J.S. gave input on and performed BLI experiments. L.F.S., Y.Z., and B.Z.H. analyzed the data. L.F.S. and B.Z.H. wrote the manuscript with edits from all co-authors.

## Competing interests

The authors declare no competing interests.
