## [Peer Review file · Nature Communications]

Divergence in a Eukaryotic Transcription Factor's co-TF Dependence Involves Multiple Intrinsically Disordered Regions

Corresponding Author: Dr Bin He

Version 0:

Reviewer comments:

Reviewer #1

(Remarks to the Author)

Snyder et al. conducted an extensive (if not exhaustive) dissection of the various protein domains found in Sc and CgPho4. They identified additional activating domains on CgPho4 and most importantly were able to show that the P2ID of ScPho4 leads to a reduction in binding capacity of Pho4 in the absence of Pho2. Whereas the P2ID of CgPho4 causes this TF to be largely independent of Pho2. This solves a long-standing question in the field and provides significant new insights into our understanding of how Pho4 functions. Overall the manuscript is well structured and well written. I thus recommend this manuscript to be published in Nature Communications. Below I provide some relatively minor suggestions which the authors might want to consider addressing in a revised manuscript prior to publication:

- Fig1b is a bit confusing. There are two models but three schematics. Maybe make it clear which schematic belongs to which model and potentially add information / schematics to help figure 1b to be more readily understood.

- x-tal structures and or alpha folded structures of both Pho4 variants might be helpful in Figure 1c? (See comment at end)

- Figure 2 provides ample evidence to support the notion that ScPho4 and CgPho4 DBDs recognise the same core consensus E-box. It would be interesting to provide more evidence to support the statement that these two TFs also recognise the same flanking bases. First the authors could provide a simple basic region amino acid alignment, as this alone could already indicate that the sequence preference is the same. Second, it might be worthwhile to drill into the PBM data and filter out subsets of sequences with the same core consensus but different flanking bases and directly compare these. At the moment Fig 2B,C are too crowded to have any chance to extract that information from the plots provided.

- it appears that the affinity differences that were observed are at best minor. The fact that only two sequences show significant differences also hints at simple measurement noise (otherwise it would be difficult to explain why only two sequences show higher affinity as opposed to an affinity difference across the board for all sequences (given that the specificity is the same)). Similarly the PBMs seem to be close to parity as well. I would thus question whether affinities are indeed different.

- the section describing Fig 3 is quite clear as is the figure itself (although it does take some time to digest all the various chimeras). Purely from an interest standpoint, it would be nice to see what activation levels a normal gal4 based activation systems achieves in order to place the activation activity obtained through these simple chimeras in some context. Are these activation potentials strong / weak etc...?

- line 189 dependence should be dependent

- line 225-226 regarding DBD swap -> see previous comment regarding the somewhat questionable conclusion that CgDBD has higher affinity

- Fig 5B-C in addition to the observed enhancing effects could the observed epistasis be simply a function of using contiguous sequences which lead to better / more accurate folding and thus function of these protein regions (understanding

that they are IDRs but maybe they are less disordered when bound to Pho2 for example)?

- line 269: wouldn't it be more accurate to say the P2ID:Sc leads to a requirement of Pho2 for activity as opposed to saying that it allows to take advantage of Pho2. The latter seems to imply to activity is high without Pho2 but can get higher with Pho2. But it seems to be the case that with P2OD:Sc the activity without Pho2 is low and becomes nominal with Pho2 present.

- the authors might reconsider calling the P2ID domain a double edged sword. At least I fail to see how it is a "double edged sword". Seems to be a standard case of an activating protein interaction.

- a schematic that graphically explains how P2ID:Pho2 and the basic region interact would be nice to have imo. Especially since this seems to be telling a very nice and interesting story and enhances our understanding of how ScPho4 functions

- I just see that the structures are shown in Fig S1. See above comment regarding structures and either ignore or move structures into main figure? It might be interesting to attempt co-folding with Pho2? Not sure what the chances are that this works (or to assess whether it worked or did not work), but should be quick to try...

Reviewer #2

(Remarks to the Author)

In this manuscript He et al. aim to reveal the role of the disordered regions in regulation of the *Saccharomyces cerevisiae* and *Candida glabrata* Pho4 transcription factors. After having tested the role of the disordered regions, the authors aimed to reveal the role of the Pho4 disordered regions in Pho2-mediated transcription. The authors provide an impressive set of experiments. However, in its present form, it was difficult for me to follow the line of argumentation and some of the statements/conclusions are far-fetched or over-interpreted in my opinion.

@Title and throughout the manuscript: the authors raise the impression that they reveal the evolution of co-dependency. In my opinion this is an overstatement as the evolutionary aspects are not covered in depth in this manuscript. Analyses of 2 Pho4 members without a thorough analysis of the exact regions involved in the interactions is not representative for an evolutionary analysis in my opinion. Moreover, an evolutionary analysis of Pho4 sequences in different species is missing in this manuscript.

Throughout the manuscript: the nomenclature used for the different sequences is highly confusing. To increase the readability, it would be helpful if the authors could use a self-explanatory and clear nomenclature. I had to go back and forth to different figures several times to find out which sequences the authors actually used for their experiments.

Constructs: the boundaries for the constructs are unclear and the definition of regions are inconsistent in the manuscript. For example, the NLS is defined as the entire region of the 3rd segment (Fig. 4), and in other figures only as a sub-region of this segment (Fig. S2). Moreover, it took me quite some time to find the boundaries for the segments shown in Fig. 4A and elsewhere (I assume that they are indicated by the label in FigS2 ?).

Results related to "CgPho4 binds the same consensus motif with a higher affinity compared to ScPho4" (from line 126): in my opinion, the results are preliminary and don't support the conclusion. First, the BLI data (Fig.2D) is not at a stage to draw a clear conclusion. Some experiments have been carried out in duplicates and statistical tests are presented for those. At least an N of 3 needs to be provided. Moreover, the data shown in Fig. S3 doesn't show a strong difference between the Sc and Cg DBD binding to DNA either. Please check the precision in Fig. S3 B&D, I assume that the data has not been obtained with a precision of 0.001 nM?

Results related to "CgPho4 encodes two additional Activation Enhancer Domains (AEDs)" (from line 156): in this paragraph, where the authors rely on computational prediction of the TAD, most of the conclusions and discussions seem to build on the assignment of the TAD to different parts of the sequence. It is unclear to me why the authors, based on their data assigned the CgPho4 TAD to a different region compared to ScPho4 TAD. The signal in the PADDLE plot (Fig 3B) is much higher for the region following CgPho4 residue 93, and the sequence in this region aligns with ScPho4. In line with this, the authors see activity with CgAD in Fig. 3C, but not the E2 region (again, confusing nomenclature).

Figure 3A and B: why are the predictions for the N/C-termini missing?

Figure 3C and D: please show the data points and provide the number of replicates

Constructs tested in Figure 3C and D: please provide a PADDLE prediction for the constructs to check if any artificial TADs are introduced by fusing different sequences

Results related to "The two Activation Enhancer Domains (AEDs) in CgPho4 increased the activity of the chimeric TFs but were insufficient to make them Pho2-independent" (from line 211): after having worked on Pho4, the authors include Pho2. Although I understand that Sc Pho4 relies on Pho2 for full activity, this and the following chapter appear disconnected from the chapters focusing on Pho4. This is mainly due to the fact that the interaction sites between Pho4 and Pho2 are left out. It is unclear which parts of Pho4 interact with Pho2 and how generation of the constructs impacts Pho2 binding. To better integrate this part, a detailed binding site mapping is required in my opinion, and a validation of the interactions using mutations of key residues in the interface as controls.

Moreover, the line of argumentation related to the models in the field are unclear to me.

Reviewer #3

(Remarks to the Author)

Snyder and colleagues study the evolution of Pho2 dependence in ScPho4 (more specifically the loss of this dependence in the *Candida glabrata* lineage). The Pho4 protein can be loosely separated in 5 functional units, but contains the canonical DNA-binding domain (DBD) and trans-activating domain (TAD) of transcription factors. They test the DBD and the TADs

independently in a foreign context (yeast 1-hybrid and in vitro studies), and perform a series of chimeras to decipher the regions that may have led to the new CgPho4 function. As always with these experiments, the results aren't simple. There exists a rich interplay between the domains that make conclusive results hard to decipher. Their main results show, however, that the DNA binding domain's intrinsic properties aren't materially different between the two orthologs, but that CgPho4 has higher trans-activating potential due to 'activator enhancing regions', and that its Pho2 interacting domain no longer 'works'.

The challenge with this study is that none of the domains, on their own, explain the complex behavior they are studying: The domains are all necessary, but not sufficient. This leads to a nagging feeling that some of the results are artifacts of the experimental design. However, overall, the authors have completed a very brave set of experiments that shed some additional lights on transcription factor and molecular evolution as a whole. Most noteworthy is the careful documentation of molecular evolution for this example.

Here are a few potential concerns:

1. Figure 1B is very confusing. The figure implies two different models, but has 3 schematics with 'vs' in between, which suggests 3 different models. I would suggest trying to think of a simpler way of displaying both models. A model of cooperative binding might be better represented by showing that Pho4 or Pho2 are not as frequently seen alone on DNA vs together, while the autoinhibition model might begin with a 'folded' Pho4 which is 'unfolded' by Pho2, unmasking its AD or its DBD. Further, the authors also test another model (increased transactivation potential of CgPho4), which doesn't seem to really show up in these figures. I also find that there might be another model which is that ScPho4 is simply not 'strong enough' without ScPho2 and that the CgPho4 enhancers substitute for ScPho2.
2. The one-hybrid assay fusing activation domains to the Gal4 DBD is used to verify the strength of various putative ADs. One potential concern here is that the activation domains can dimerize to the endogenous Pho2/Pho4 of the yeast strain. Could this potentially obscure the interpretation of what's happening? This is especially interesting for the E1/E2 fusions. I am also not aware of a mechanism for 'activation enhancer domains'. This is not something I can find in the literature, so I am not sure what they are. Do they work in Trans? Do they recruit something?
3. For Figure 5, Is this triangular heatmap necessary? What is the reason for not displaying all the 32 constructs as a bar chart? The 'scale' is also confusing here. The heat scale goes from -20 to 20 for ScPho4 in the presence of Pho2. Presumably it is the same scale for the bottom triangle as well, so it's the activity in the absence of Pho2. However, in the bar chart of Figure 5b, the numerical values are relative to ScPho4 (in the presence and absence of Pho2, respectively). Can the authors standardize the scales so things are comparable? Otherwise it's impossible to tell what a +10 activity is (2 fold or 1.01 fold?)
4. It seems to me that the authors map the 'auto-inhibitory' region to the P2ID region next to the DBD. However, the mechanism for how this works doesn't seem to be explored. Does the P2ID prevent binding of the DBD in the absence of Pho2 or does it block the activation domains? One of the striking thing about this P2ID region in CgPho4 is that it is severely elongated. This deletion in ScPho4 removes a potential basic-rich IDR (usually nuclear localization signals but here it could be something else), but more importantly moves the Pho85 motif very close to the DBD. This Pho85 motif is known to control the Pho4-Pho2 interaction, so we can presume that Pho2 binds to that section. Based on this, it is possible that the P2ID of ScPho4 blocks the DBD in the absence of Pho2. Can't the authors show this?
5. One of the problems when making chimeras by swapping IDRs of different lengths, is that the separation between functional elements is now perturbed. It is usual in the literature to try to minimize this confounding. It's difficult here to assess whether this has played a role in some of the 'surprising' results. For example, but the NLS and the P2ID are very large in CgPho4 and splits conserved elements into different spatial locations. One way this has been studied in the literature is to scrambled versions of the non-conserved elements, swap the conserved elements, and test to see if the distance matters. Of course, none of this may solve the potential fact that the breakpoints for making chimeras might not be correct. It seems the breakpoints were chosen to be well within conserved mini-blocks, which I find to be an odd choice. Finally, it is usual in this field to interpret positive results of chimeras, but be less assertive with negative results because it is never clear if a negative result is due to the design itself. This is why adding a tiny IDR to the DBD seems to completely change results and so one should be very careful making negative assertions.
6. One of the possibly puzzling result from this example is that there are some transcripts that DO require CgPho2 for binding. In He et al (2017), the authors showed that ~14 CgPho4 ChIP peaks require CgPho2. What makes them special? Are the recognition sites different? I would recommend the authors use those promoters as a control for their study here rather than use a single promoter.

Here are some minor comments:

1. Line 159, CNNs are 'convolutional' neural networks, not 'convoluted'.
2. Line 294, typo: on 'their' own.
3. The 'nuclear fraction' calculation does not make any sense to me. It does not correct for the size of the nucleus, which can be done by averaging over the area.
4. The proteins are purified with a GST tag, but GST is known to dimerize. Does this interfere with their measurements?

Version 1:

Reviewer comments:

Reviewer #1

(Remarks to the Author)

The authors satisfactorily addressed all of my comments in their revised version.

Reviewer #2

(Remarks to the Author)

The authors have addressed my earlier comments.

Reviewer #3

(Remarks to the Author)

The authors have adequately addressed my previous concerns and I can recommend this manuscript for publication.

REVIEWER COMMENTS

Reviewer #1 (Remarks to the Author):

Snyder et al. conducted an extensive (if not exhaustive) dissection of the various protein domains found in Sc and CgPho4. They identified additional activating domains on CgPho4 and most importantly were able to show that the P2ID of ScPho4 leads to a reduction in binding capacity of Pho4 in the absence of Pho2. Whereas the P2ID of CgPho4 causes this TF to be largely independent of Pho2. This solves a long-standing question in the field and provides significant new insights into our understanding of how Pho4 functions. Overall the manuscript is well structured and well written. I thus recommend this manuscript to be published in Nature Communications. Below I provide some relatively minor suggestions which the authors might want to consider addressing in a revised manuscript prior to publication:

We appreciate the reviewer's recognition of the importance of our work.

- Fig1b is a bit confusing. There are two models but three schematics. Maybe make it clear which schematic belongs to which model and potentially add information / schematics to help figure 1b to be more readily understood.

We thank the reviewer for the suggestion. We have now revised Fig. 1b to have schematics clearly associated with each of the two models.

- x-tal structures and or alpha folded structures of both Pho4 variants might be helpful in Figure 1c? (See comment at end)

We thank the reviewer for the question. The AlphaFold2 predicted structures were included in Figure S1 and referred to in the text after describing the domain architecture. We added the following sentence (line 119-120)

"Both ScPho4 and CgPho4 were predicted to be mostly intrinsically disordered outside the DBD, with some low confidence helices in R1 and AD (Fig. S1)."

- Figure 2 provides ample evidence to support the notion that ScPho4 and CgPho4 DBDs recognise the same core consensus E-box. It would be interesting to provide more evidence to support the statement that these two TFs also recognise the same flanking bases. First the authors could provide a simple basic region amino acid alignment, as this alone could already indicate that the sequence preference is the same. Second, it might be worthwhile to drill into the PBM data and filter out subsets of sequences with the same core consensus but different flanking bases and directly compare these. At the moment Fig 2B,C are too crowded to have any chance to extract that information from the plots provided.

We thank the reviewer for the great suggestions. We now include a bHLH domain alignment as Fig. 2A, highlighting the residues known to recognize nucleotide bases and phosphate backbones based on the crystal structure for ScPho4 DBD with its cognizant DNA (PDB: 1A0A). As for the PBM data, we agree with the reviewer that the information is too crowded

to be useful. The purpose of the PBM data is to show that (1) a comprehensive scan on the 9-mer space reveals no difference in binding preference; (2) complementary to the universal 9-mer library, a 36-mer selected library was used to examine binding preference with longer flanking nucleotide sequences. Again, no obvious difference was revealed. For the first point, it is appropriate to show *all* the data. However, we now use red colors to highlight the 12 oligos included in the library that contain the consensus E-box motif. To illustrate the second point, we followed the reviewer's suggestion to present the gcPBM data in two sets based on whether the 36-mer contains the consensus motif or not. This way, we highlight any potential preference difference due to the flanking bases. They are now presented as Fig. S3.

- it appears that the affinity differences that were observed are at best minor. The fact that only two sequences show significant differences also hints at simple measurement noise (otherwise it would be difficult to explain why only two sequences show higher affinity as opposed to an affinity difference across the board for all sequences (given that the specificity is the same)). Similarly the PBMs seem to be close to parity as well. I would thus question whether affinities are indeed different.

We thank the reviewer for raising this point. Indeed, we noted in the manuscript that the affinity difference is minor (Lines 333 and onwards in the original submission), and the genetic dissection also didn't support a functional contribution from the DBD to reduced Pho2-dependence (Fig. 5). We concluded that there is no evidence for binding affinity as a main contributor to the Pho2-dependence difference between the two Pho4 orthologs.

Regarding the reviewer's concern about the observed difference likely due to noise, we believe our data supports a reproducible, 3-fold difference in affinity between the two Pho4 orthologs' DBDs against the consensus motif. This is based on the Biolayer Interferometry experiments for the consensus motif, which was repeated 5 times, utilizing three batches of protein preps made on different days, all supporting the same difference. We now present this data more clearly in Fig. 2D, E. Moreover, we validated the results using an independent method, i.e., EMSA, which also supported the conclusion (Fig. S4).

The reviewer raised concerns over the finding that only two of the ten oligos tested showed a significant difference in binding. We think this is not unexpected based on how DBDs recognize DNA. It is known that DBD specificity is mediated by a network of amino acid to nucleotide base and amino acid to DNA backbone contacts. It is reasonable to anticipate that changes at one or more of those amino acid positions can specifically affect some but not all positions in a motif and thus affect the affinity for different sequence variants.

In summary, we agree with the reviewer that the difference in affinity between the two orthologs are modest at best. We added the following sentences in the Results (lines 162-

165) to clarify this point *“It is worth noting that this observed difference in affinity, while statistically significant, is modest ($\Delta\Delta G = -0.87$ kcal/mol, based on the mean BLI measurements, assuming a temperature of 25°C). Its potential impact on gene regulation depends critically on the effective concentration of Pho4 in the nucleus.”* Also, we note that the affinity results for the nine one-bp-off variants do not contribute to our main conclusions. Therefore, in the revised version we only retain the BLI results against the consensus motif, which allows us to show both the binding curves and the K_D estimates (revised Fig. 2).

- the section describing Fig 3 is quite clear as is the figure itself (although it does take some time to digest all the various chimeras). Purely from an interest standpoint, it would be nice to see what activation levels a normal gal4 based activation systems achieves in order to place the activation activity obtained through these simple chimeras in some context. Are these activation potentials strong / weak etc...?

We thank the reviewer for the suggestion. We included the full length Gal4 as a positive control in our original experiments. We didn't include it in the figure because doing so would compress the scale, as its activity is higher than the experimental constructs. Below is all the raw data plotted, including Gal4 full length. Note that the experiment was repeated on two different days. The results were highly consistent. We presented one of the two sets of data in the main figure. Conclusions stay the same whether we use the other date or both.

Fig. R1 Yeast One-hybrid data with full length Gal4. Figure formats are the same as Fig. 3

- line 189 dependence should be dependent
We have corrected this. Thanks!

- line 225-226 regarding DBD swap -> see previous comment regarding the somewhat questionable conclusion that CgDBD has higher affinity

We thank the reviewer for this comment. As we explained in our response to a previous question, our data showed conclusively that there is a modest but reproducible difference in the binding affinity towards the consensus motif between the two Pho4 DBDs. Despite the higher affinity, swapping the DBD alone didn't rescue the Pho2-dependence of ScPho4. We discussed the implications of this unexpected finding (lines 333 and onwards in the original submission).

- Fig 5B-C in addition to the observed enhancing effects could the observed epistasis be simply a function of using contiguous sequences which lead to better / more accurate folding and thus function of these protein regions (understanding that they are IDRs but maybe they are less disordered when bound to Pho2 for example)?

We thank the reviewer for the suggestion. When designing breakpoints, we avoided regions with predicted alpha helices to minimize the artifacts when breaking and fusing the parts (Fig. S1, S2). It is also worth noting that the regions exhibiting epistasis, i.e., regions 1 to 3 (R1, AD and NLS), do not mediate Pho2-interaction in ScPho4. The region interacting with Pho2 has been mapped to the IDR next to the DBD, which we refer to as P2ID in this study. Our yeast two-hybrid data further suggest that CgPho4 doesn't interact with Pho2 from either species efficiently (Fig. 6D). Despite these, we acknowledge that one cannot rule out the possibility of the observed epistasis effects being a result of the use of contiguous sequences leading to better or more accurate folding. We now include this explanation in our discussion of the results (lines 258-263):

"One explanation for the observed epistasis may be that the two CgPho4 AEDs work more efficiently with AD:Cg than with AD:Sc. However, it could also be explained by the disruption of the native conformation and function of the "interacting" regions resulting in a lower activity in the species-mixed constructs. Note that to minimize such effects, we designed the breakpoints to avoid any predicted secondary structures (Fig. S1, S2)."

- line 269: wouldn't it be more accurate to say the P2ID:Sc leads to a requirement of Pho2 for activity as opposed to saying that it allows to take advantage of Pho2. The latter seems to imply to activity is high without Pho2 but can get higher with Pho2. But it seems to be the case that with P2OD:Sc the activity without Pho2 is low and becomes nominal with Pho2 present.

We thank the reviewer for this question. The basis for our conclusion in line 269 was from the observation that when swapping P2ID:Cg in CgPho4 for the corresponding P2ID:Sc, we observed a higher A_{PHO2} (than CgPho4 itself) and a dramatically reduced $A_{\text{pho2}\Delta}$ (Fig. 6B, row 3). The former led us to wonder if P2ID:Sc functions to enhance the TF activity only when Pho2 is present. This is then confirmed when we split the P2ID:Sc and found that the first half of P2ID:Sc has a *purely positive effect* on TF function with or without Pho2 in the CgPho4 background (Fig. 6B, row5). Nonetheless, we agree with the reviewer that the quoted line could cause confusions. We thus revised this section as follows (lines 288-290): “Conversely, swapping ScPho4’s P2ID (P2ID:Sc) into CgPho4 increased A_{PHO2} beyond that of CgPho4 (18 to 22) but reduced $A_{\text{pho2}\Delta}$ (17 to 4.8). This suggests that P2ID:Sc is a key factor to Pho2-dependence.”

- the authors might reconsider calling the P2ID domain a double edged sword. At least I fail to see how it is a “double edged sword”. Seems to be a standard case of an activating protein interaction.

We thank the reviewer for this question. As we mentioned in our response to the previous question, our main rationale for this interpretation comes from our observation that the N-terminal half of the P2ID:Sc increases the chimeric Pho4 activity with and without Pho2 while the second half reduces the activity in both contexts. Thus, we think it is not simply an activating protein interaction, which would suggest a net positive effect, while P2ID:Sc is net negative when *pho2* is absent. We hope this satisfies the reviewer’s question.

- a schematic that graphically explains how P2ID:Pho2 and the basic region interact would be nice to have imo. Especially since this seems to be telling a very nice and interesting story and enhances our understanding of how ScPho4 functions

We thank the reviewer for the suggestion and appreciation of our work’s contribution. We now followed the suggestion and added a model summary (Fig. 7), in which we included a cartoon demonstrating what we learned about the function of the P2ID:Sc and its effect on ScPho4’s dependence on Pho2.

- I just see that the structures are shown in Fig S1. See above comment regarding structures and either ignore or move structures into main figure? It might be interesting to attempt co-folding with Pho2? Not sure what the chances are that this works (or to assess whether it worked or did not work), but should be quick to try...

We thank the reviewer for this suggestion. We were also interested to see if the newly released AlphaFold 3 could give us insight into the Pho4-Pho2 interaction. We performed the prediction with two ScPho4 molecules (known to function as a dimer) and one ScPho2 molecule. However, the confidence of the predicted interface was very low. According to

the AlphaFold3 documentation, an ipTM score > 0.8 is considered high confidence, between 0.6 and 0.8 is considered the gray zone, and lower than 0.6 considered a failed prediction. The ipTM score for ScPho4-ScPho2 was 0.34. That being said, the prediction interface aligns with regions in both ScPho4 and ScPho2 implicated in their interaction (Fig. R2, Hirst et al. 1994, PMID: 8187772, Bhoite et al. 2002, PMID: 12145299). The predicted interactions between CgPho4 and both ScPho2 and CgPho2 had even lower scores and therefore are not presented here. Overall, we find the prediction by AF3 to be both promising and still low-confidence for providing mechanistic insight into the TF-TF interaction and its evolution.

Reviewer #2 (Remarks to the Author):

In this manuscript He et al. aim to reveal the role of the disordered regions in regulation of the *Saccharomyces cerevisiae* and *Candida glabrata* Pho4 transcription factors. After having tested the role of the disordered regions, the authors aimed to reveal the role of the Pho4 disordered regions in Pho2-mediated transcription. The authors provide an impressive set of experiments. However, in its present form, it was difficult for me to follow the line of argumentation and some of the statements/conclusions are far-fetched or over-interpreted in my opinion.

We appreciate the reviewer's recognition of our work and the critical comments, which we will address below.

@Title and throughout the manuscript: the authors raise the impression that they reveal the evolution of co-dependency. In my opinion this is an overstatement as the evolutionary aspects are not covered in depth in this manuscript. Analyses of 2 Pho4 members without a thorough analysis of the exact regions involved in the interactions is not representative for an evolutionary analysis in my opinion. Moreover, an evolutionary analysis of Pho4 sequences in different species is missing in this manuscript.

We appreciate the reviewer's criticism. We can see why the reviewer raised this criticism, in that examining the divergence between two orthologous TFs doesn't allow us to reveal the *general pattern* of how Pho4's dependence on Pho2 evolved. In our original title, we used the word "Evolution" to mean evolved differences. However, the word often implies more than two species/orthologs. To avoid the appearance of overstating what our study did, we now revised the title to read "~~Evolution~~ Divergence in a Eukaryotic Transcription Factor's co-TF Dependence Involves Multiple Intrinsically Disordered Regions Affecting Activation and Autoinhibition".

The reviewer mentioned the "exact regions involved in the interactions" not being revealed. If the reviewer was referring to the regions involved in the interaction between Pho4 and Pho2, then it was previously identified for ScPho4. The region involved, aa 200-218 in ScPho4, was part of the P2ID region in the present study. It includes one of the phosphorylatable serines targeted by the Pho85/80 kinase complex, which directly controls ScPho4 and ScPho2 interaction (Komeili and O'Shea 1999, PMID: 10320381). We added this background information to the first section of the Results. As for CgPho4, our yeast 2-hybrid assay didn't detect any physical interaction between it with Pho2 (Fig. 6).

Lastly, we fully agreed with the reviewer's suggestion and performed evolutionary analyses of Pho4 sequences in an extended group of species. Specifically, we chose species based on our prior Pho2-dependence test in the *S. cerevisiae* common background (He et al. 2017)

to include both species whose Pho4 orthologs exhibited reduced Pho2-dependence and those whose Pho4 orthologs depended on Pho2, while spanning ~200 million years of divergence. The result showed a few intriguing patterns. First, Pho4 orthologs that are less dependent on Pho2 were predicted to have additional regions with activation potential (AED candidates) outside the main activation domain while all but one Pho2-dependent Pho4 according to our test did (Fig. X8, *N. bacillisporus*). Notably, the distantly related *C. albicans* doesn't require Pho2 for PHO gene induction, and its Pho4 exhibited two distinct regions with predicted activation potential, similar to that of CgPho4. Second, we found that P2ID lengths are generally longer among Pho4 orthologs in the *Nakaseomyces* genus compared with *S. cerevisiae* and its close relative *S. mikatae* as well as in the outgroup, *L. kluyveri*. While the correlation is not perfect for either feature, we are encouraged by this observation, which suggests to us that the two mechanisms behind the divergence in Pho2 dependence between ScPho4 and CgPho4 may be more generally at play in the evolution of Pho4. We now included this result as Fig. S8 and discussed its implication in the Discussion.

Throughout the manuscript: the nomenclature used for the different sequences is highly confusing. To increase the readability, it would be helpful if the authors could use a self-explanatory and clear nomenclature. I had to go back and forth to different figures several times to find out which sequences the authors actually used for their experiments. Constructs: the boundaries for the constructs are unclear and the definition of regions are inconsistent in the manuscript. For example, the NLS is defined as the entire region of the 3rd segment (Fig. 4), and in other figures only as a sub-region of this segment (Fig. S2). Moreover, it took me quite some time to find the boundaries for the segments shown in Fig. 4A and elsewhere (I assume that they are indicated by the label in FigS2 ?).

We appreciate the reviewer's criticism and suggestions. We made the following edits to improve the consistency and clarity of the region nomenclature: in Fig. S2, we now use the same boundaries as in Fig. 1 and 4. As a result, we now consistently refer to the five regions as R1, AD, NLS, P2ID and DBD throughout the study. The only exception is In Figure 3, where we used a different set of boundaries to specifically test the predictions of the activation potential by PADDLE. To help the reader easily identify the boundaries for all the different regions tested, we now include a table (Table 1) with the start and end positions for all segments tested in the paper.

Results related to "CgPho4 binds the same consensus motif with a higher affinity compared to ScPho4" (from line 126): in my opinion, the results are preliminary and don't support the conclusion. First, the BLI data (Fig.2D) is not at a stage to draw a clear conclusion. Some experiments have been carried out in duplicates and statistical tests are presented for those. At least an N of 3 needs to be provided.

We appreciate the reviewer's concern. All BLI experiments had at least three replicates. The consensus motif "CACGTG" was measured 5 times using separate batches of protein prep. The difference in K_D between the two Pho4 DBDs with respect to the consensus motif was highly reproducible. (also see our response to the previous reviewer's comment similar to this one). Also as we explained in a previous response, because the non-consensus motif results do not add to our main conclusion, we decided to present the consensus motif result alone, along with representative fractional binding curves used to derive the K_D estimates to help the readers assess the strength of the evidence. Statistical tests, including a Student's t-test (on log transformed K_D) and a non-parametric Wilcoxon rank-sum test gave the same conclusion, that is, there is a significant, ~3-4 fold difference in K_D between the two Pho4 DBDs. This was independently validated by a different technique, i.e., EMSA (gel shift, Fig. S4).

Moreover, the data shown in Fig. S3 doesn't show a strong difference between the Sc and Cg DBD binding to DNA either. Please check the precision in Fig. S3 B&D, I assume that the data has not been obtained with a precision of 0.001 nM?

We appreciate the reviewer's concern. We acknowledged above that the difference in K_D is modest, which is why the gel shift result may not appear obviously different at first sight. With respect to the precision of the K_D estimates, the reviewer was correct in that we didn't vary the protein concentrations at 0.001 nM increments. The reason why the K_D estimates were expressed with three decimal points was because it was obtained by fitting a one-site specific binding curve with Hill coefficient to the measured fractional-bound vs protein concentration measurements. That said, we think it is not necessary or useful to show the estimates with more than one decimal point, since that's sufficient to demonstrate the

difference between the two Pho4 DBDs. Therefore, we have modified Fig S4 to present K_D with one decimal point precision.

Results related to “CgPho4 encodes two additional Activation Enhancer Domains (AEDs)” (from line 156): in this paragraph, where the authors rely on computational prediction of the TAD, most of the conclusions and discussions seem to build on the assignment of the TAD to different parts of the sequence. It is unclear to me why the authors, based on their data assigned the CgPho4 TAD to a different region compared to ScPho4 TAD. The signal in the PADDLE plot (Fig 3B) is much higher for the region following CgPho4 residue 93, and the sequence in this region aligns with ScPho4. In line with this, the authors see activity with CgAD in Fig. 3C, but not the E2 region (again, confusing nomenclature).

Figure 3A and B: why are the predictions for the N/C-termini missing?

Figure 3C and D: please show the data points and provide the number of replicates

We appreciate the reviewer’s comments on Figure 3. Regarding the TAD assignment, we believe the reviewer was referring to the 9aaTAD motif in CgPho4 being detected outside of the region aligned with the ScPho4 AD. Indeed, as the reviewer pointed out, our Y1H data supported that region of CgPho4 as the main activation domain. The confusion comes from the name “9aaTAD”, which refers to a 9 aa motif and not the activation domain per se. This motif was initially identified based on *S. cerevisiae* studies (Piskacek *et al.* 2007, PMID: 17467953). The presence of a match to this motif is often, but not always, predictive of activation potential. In our case, the best match to the 9aaTAD motif in CgPho4 happens to lie not in the main AD, but in the second AED (E2). To avoid the confusion between 9aaTAD and the functional AD, we removed the label “9aaTAD” from Fig. 3A, B. We kept the orange triangles indicating the predicted motif matches and the name is mentioned in the legend.

Fig. 3A and B: the reason that the N/C-termini data are missing in the prediction is because PADDLE makes predictions on 53 aa blocks. A sliding window with 1 aa increment was used to obtain estimates throughout the protein. The predicted Z-score for each 53 aa block was plotted against the middle position of the block, resulting in the first and last 26 aa with no values. This is explained in the PADDLE paper (Sanborn *et al.* 2021 eLife). We have now included a short explanation in the legend

“Predictions were made in 53 aa blocks and plotted using the middle of each block as the X coordinate. As a result, the first and last 26 aa positions have no scores.”

Fig. 3C/D: We thank the reviewer for the suggestion. We have revised the figure to show individual data points and noted the number of replicates on the plot. The same changes were made to all bar plots in the manuscript (Figs. 5 and 6).

Constructs tested in Figure 3C and D: please provide a PADDLE prediction for the constructs to check if any artificial TADs are introduced by fusing different sequences

We appreciate the reviewer's concern and suggestion. We performed PADDLE prediction for all constructs used in Figures 3C and D. The results showed that 1) the Gal4 DBD common to all constructs doesn't possess activation potential on its own; 2) no aberrant activation potential peaks emerged as a result of artificially fusing regions (e.g., no signals near the boundaries the Gal4DBD and the tested Pho4 regions). The PADDLE prediction is included as Fig. S5.

Results related to "The two Activation Enhancer Domains (AEDs) in CgPho4 increased the activity of the chimeric TFs but were insufficient to make them Pho2-independent" (from line 211): after having worked on Pho4, the authors include Pho2. Although I understand that Sc Pho4 relies on Pho2 for full activity, this and the following chapter appear disconnected from the chapters focusing on Pho4. This is mainly due to the fact that the interaction sites between Pho4 and Pho2 are left out. It is unclear which parts of Pho4 interact with Pho2 and how generation of the constructs impacts Pho2 binding. To better integrate this part, a detailed binding site mapping is required in my opinion, and a validation of the interactions using mutations of key residues in the interface as controls.

We thank the reviewer for making this comment and suggestion. We would first like to clarify the extent of knowledge already present on the ScPho4-ScPho2 interaction. A study from the O'Shea lab showed that phosphorylation of the SP6 site by Pho85 (within the P2ID region, Fig. S2) of ScPho4 blocked the interaction with Pho2, indicating that this region is involved in the interaction (Komeili and O'Shea, 1999, PMID: 10320381). Another study used truncation and internal deletion mutants of ScPho4 to narrow down the required residues for interaction to aa200-218 (Hirst et al, 1994, PMID: 7957107). We have added this to the first Result section where we described domain organizations and sequence properties of ScPho4 and CgPho4. Additionally, our yeast 2 hybrid results show that CgPho4 has low to no interaction with Pho2. Therefore, while we appreciate the reviewer's suggestion to map the residues involved in Pho4-Pho2 interactions, we think this is in part done for ScPho4 and not relevant for CgPho4. We also believe, given our main results, that the divergence in Pho2 dependence between ScPho4 and CgPho4 doesn't primarily result from the gain or loss of Pho2-interactions (it can be a consequence of becoming less dependent on Pho2 as seen in CgPho4).

Moreover, the line of argumentation related to the models in the field are unclear to me.

We thank the reviewer for pointing out the lack of clarity in explaining existing models. We assume the reviewer was referring to the description in the introduction. We have now revised the model figure (Fig. 1) to clearly distinguish the two non-mutually exclusive models. Model I stipulates that CgPho4 has stronger DNA binding and transactivation abilities, allowing it to function without the co-TF Pho2. The second model assumes that ScPho4 activity is both boosted when Pho2 is present and autoinhibited without Pho2. Our

data provided support for both models, although the chimeric Pho4 reporter assays argued against a classic “key-and-lock” autoinhibition model because if that were to be the case, we would expect at least one of the chimeric constructs show a significant gain of activity without Pho2 by swapping out the “lock” part within ScPho4. Instead, our results suggested an intriguing explanation where the “lock” in ScPho4, localized to the P2ID region, is required for the TF’s function, such that breaking the lock also breaks the TF as a whole (Fig. 6C). We hope the above changes and explanation help answer the reviewer’s question.

Reviewer #3 (Remarks to the Author):

Snyder and colleagues study the evolution of Pho2 dependence in ScPho4 (more specifically the loss of this dependence in the *Candida glabrata* lineage). The Pho4 protein can be loosely separated in 5 functional units, but contains the canonical DNA-binding domain (DBD) and trans-activating domain (TAD) of transcription factors. They test the DBD and the TADs independently in a foreign context (yeast 1-hybrid and *in vitro* studies), and perform a series of chimeras to decipher the regions that may have led to the new CgPho4 function. As always with these experiments, the results aren’t simple. There exists a rich interplay between the domains that make conclusive results hard to decipher. Their main results show, however, that the DNA binding domain’s intrinsic properties aren’t materially different between the two orthologs, but that CgPho4 has higher trans-activating potential due to ‘activator enhancing regions’, and that it’s Pho2 interacting domain no longer ‘works’. The challenge with this study is that none of the domains, on their own, explain the complex behavior they are studying: The domains are all necessary, but not sufficient. This leads to a nagging feeling that some of the results are artifacts of the experimental design.

However, overall, the authors have completed a very brave set of experiments that shed some additional lights on transcription factor and molecular evolution as a whole. Most noteworthy is the careful documentation of molecular evolution for this example.

We appreciate the reviewer’s comments both acknowledging the scale of the work and the challenges in solving protein divergence mysteries, especially when single domain changes are insufficient to explain the difference. We are aware of these challenges, but also believe that divergence in TF proteins hold the “missing” key to the evolution of gene regulation. We believe our results both provided novel insights and raised significant questions to be answered with further studies.

We think the “no single region explains the mystery on its own” is a valuable observation if looked at from another perspective: it shows that at the same time when protein domains may exhibit modularity, e.g., DNA binding domains and activation domains can work out of context, the full functionality of a TF, when assessed in its *in vivo* context, often features epistatic interactions between regions. We mentioned this in our Discussion.

Nonetheless, we understand why the reviewer had the nagging feeling as mentioned above. Our response to this general comment, which we will elaborate in our specific responses below, is that we were reassured by the agreement between the results we obtained for individual regions “out of context”, e.g., activation potential predictions and measurements by Y1H, and our chimeric Pho4 experiments measuring their effect in the context of the full length Pho4 inside the cell. We hope to convince the reviewer with our responses below that our major conclusions are well supported by our data.

Here are a few potential concerns:

1. Figure 1B is very confusing. The figure implies two different models, but has 3 schematics with ‘vs’ in between, which suggests 3 different models. I would suggest trying to think of a simpler way of displaying both models. A model of cooperative binding might be better represented by showing that Pho4 or Pho2 are not as frequently seen alone on DNA vs together, while the autoinhibition model might begin with a ‘folded’ Pho4 which is ‘unfolded’ by Pho2, unmasking its AD or its DBD. Further, the authors also test another model (increased transactivation potential of CgPho4), which doesn’t seem to really show up in these figures. I also find that there might be another model which is that ScPho4 is simply not ‘strong enough’ without ScPho2 and that the CgPho4 enhancers substitute for ScPho2.

We thank the reviewer for pointing out the confusion and providing specific suggestions. We have now revised the model figure, where we eliminated the “vs”, and used boxes to clearly delineate the two models. We followed the reviewer’s suggestion to present ScPho4 in the “folded” vs “open” state, with a “+Pho2” in the middle indicating the effect of the co-TF. To illustrate the “enhanced activation potential” hypothesis that we tested in the paper, we now added an “asterisk” to the AD and DBD in the CgPho4 enhanced activity model.

The reviewer suggested a third model, where “ScPho4 is simply not ‘strong enough’”. We think this is equivalent to our first, “enhanced activity” model, since both posit that the difference in co-TF dependence stems from the activity difference between the two Pho4 orthologs – ScPho4 is weaker is the same as saying CgPho4 is stronger. Therefore, we stayed with the two models in the manuscript.

2. The one-hybrid assay fusing activation domains to the Gal4 DBD is used to verify the strength of various putative ADs. One potential concern here is that the activation domains can dimerize to the endogenous Pho2/Pho4 of the yeast strain. Could this potentially obscure the interpretation of what’s happening? This is especially interesting for the E1/E2 fusions. I am also not aware of a mechanism for ‘activation enhancer domains’. This is not something I can find in the literature, so I am not sure what they are. Do they work in Trans? Do they recruit something?

We thank the reviewer for raising this question. First of all, the endogenous Pho4 is not active in the condition we performed the Y1H experiment: the growth media contains 7.5 mM phosphate, at which level the endogenous Pho4 is phosphorylated and actively kept outside the nucleus (O'Neill *et al.* 1996, PMID: 8539622). To verify this, we tested for Pho4 activity by measuring the levels of secreted phosphatase encoded by *PHO5*, which is one of the most strongly induced genes by Pho4 (Fig. R3). As expected, the Y1H host strain without *pho2* didn't show phosphatase production when grown in either high or no phosphate conditions. The Y1H strain with *PHO2* only showed phosphatase production when grown in the no phosphate condition but not under high phosphate. This confirms that the endogenous Pho4 is not active in the condition under which the Y1H experiment was conducted (high phosphate).

Pho2 nuclear localization is not known to be regulated by phosphate availability. To test if any of the regions showing activation potential in our Y1H assay are dependent on Pho2 for their activity, we deleted *pho2* in the Y1H host strain and repeated the assay (Fig. S6).

A pairwise two-sided t-test comparing the activity of the constructs with and without Pho2 didn’t identify any construct with a significant Pho2-dependence at the 0.05 level after Bonferroni-Holm correction. While not significant, we noticed that constructs containing CgE1 showed a consistent trend of a higher activity with than without Pho2 (Fig. S6, blue vertical line). None of them reached the significance threshold after the multiple-testing correction (corrected *P*-values > 0.15). It is important to note that all three still showed a significant boosting effect compared to the respective AD alone in the absence of Pho2. For example, CgE1:CgAD and CgAD:CgE1 both showed significantly higher activity than CgAD without *pho2* (two-sided t-test *p*-values < 0.05 after Bonferroni correction). The same is true for CgE1:ScAD_9aa and ScAD_9aa:CgE1 compared with ScAD_9aa alone. Therefore, we conclude that CgE1 possesses activation enhancing activity independent of Pho2. However, this data suggests the possibility that CgE1 also possesses a weak Pho2-interaction activity even though our Yeast-2-Hybrid assay detecting no interaction between CgPho4 (excluding

DBD due to Y2H design) and Pho2 (Fig. 6). This hypothesis needs to be tested using more sensitive assays such as the split DHFR-based protein fragment complementation assay.

In summary, we conclude that CgE1 and CgE2 in CgPho4 can boost the activity of the main AD and that this activity doesn't require Pho2. We now include the figure above with a table showing t-test results as Fig. S6 and added a short description in the main text.

We agree with the reviewer that the concept of AED was not mentioned in the literature as far as we know, and therefore represents a novel discovery from our work. We speculate that the AEDs may contribute to activation by independently recruiting cofactors, hence strengthening activation. However, they alone are not sufficient to recruit all the necessary cofactors, making their activity dependent on the presence of the main AD. Alternatively, we envision that the AEDs, being directly adjacent to the AD in the protein, may allosterically influence the main AD, e.g., stabilizing its interactions with the cofactor, hence enhancing its effect. We now added a section in the Discussion to describe our hypotheses (lines 353-359):

“We are not aware of existing examples or proposed mechanisms for such a phenomenon. One hypothesis for how AEDs work is that they are weaker ADs with the same biochemical activities, i.e., recruiting cofactors through protein-protein interactions. As such, they are not sufficient for activation by themselves but can increase the activity of the nearby AD (the effect can be non-additive). Alternatively, AEDs may affect the conformation of the AD and have no activity on their own. Further tests by synthetic constructs and biochemical assays, such as co-IP followed by mass-spec, will provide mechanistic insight into how AEDs function.”

3. For Figure 5, Is this triangular heatmap necessary? What is the reason for not displaying all the 32 constructs as a bar chart? The 'scale' is also confusing here. The heat scale goes from -20 to 20 for ScPho4 in the presence of Pho2. Presumably it is the same scale for the bottom triangle as well, so it's the activity in the absence of Pho2. However, in the bar chart of Figure 5b, the numerical values are relative to ScPho4 (in the presence and absence of Pho2, respectively). Can the authors standardize the scales so things are comparable? Otherwise it's impossible to tell what a +10 activity is (2 fold or 1.01 fold?)

We thank the reviewer for the question and suggestions. We now present all chimeras on a scatter plot, with the x and y-axis values being the mean of their activity measurements with and without Pho2. We tested showing them all as bar plots and found it too crowded to be helpful. In fact, we created a shinyapp for exploratory data analysis for this reason. It allows us and anyone to select subsets of the chimeras and show the underlying measurements as well as high level summaries. The app is publicly accessible at the following URL:

<https://binhe-lab.shinyapps.io/Pho4-chimera-data-plotter-v3/>

We decided to retain one heat map on the ScPho4 background because it presents a full picture of all one and two region activity differences between CgPho4 and ScPho4, when measured in the ScPho4 background. We have modified the legend for the color scale and also the legend to help clarify that the values presented are differences in activities between the CgPho4 region and the counterpart from ScPho4, which explains why the scales are different in the heat map from that in the scatter or bar plots. To avoid confusion, we also changed the scale in the bar plot from the “relative activity against ScPho4” to their raw values to be consistent with what’s shown in the scatter plot. This way, we now have the same scales for the scatter and bar plots, while the heat map values and the linear model estimates in panel D both represent differences in activities (ΔA Cg-Sc).

4. It seems to me that the authors map the ‘auto-inhibitory’ region to the P2ID region next to the DBD. However, the mechanism for how this works doesn’t seem to be explored. Does the P2ID prevent binding of the DBD in the absence of Pho2 or does it block the activation domains? One of the striking thing about this P2ID region in CgPho4 is that it is severely elongated. This deletion in ScPho4 removes a potential basic-rich IDR (usually nuclear localization signals but here it could be something else), but more importantly moves the Pho85 motif very close to the DBD. This Pho85 motif is known to control the Pho4-Pho2 interaction, so we can presume that Pho2 binds to that section. Based on this, it is possible that the P2ID of ScPho4 blocks the DBD in the absence of Pho2. Can’t the authors show this?

We thank the reviewer for the thoughtful suggestions. To test the hypothesis that P2ID:Sc inhibits binding of the DBD:Sc in the absence of Pho2, we performed ChIP-qPCR on DBD:Sc alone and P2ID+DBD:Sc both in the presence and absence of Pho2 (Fig. R4). To our surprise, we found that while the enrichment score is much lower across a panel of target genes in the latter construct, there wasn’t a detectable difference between the with and without Pho2 background. Additional experiments are needed to validate and interpret this result. In other unpublished experiments, we found preliminary evidence that P2ID:Sc is able to suppress the activation activity of the main AD without Pho2 while boosting it with Pho2. Thus, the mechanisms of how P2ID:Sc affect the TF function is an intriguing and open question.

We also tested the hypothesis that the length of P2ID affects its autoinhibitory effect. We constructed two chimeras in which we inserted either a portion of the P2ID:Cg (aa 392-451) or an equal length (GS)x30 linker in between P2ID:Sc and DBD:Sc (between aa 235 and 236) to move the former further away from the latter. Our results showed that simply increasing the distance between P2ID and DBD didn't affect the Pho2-dependence of ScPho4 (Fig. R5). Moreover, our original chimera set included ones where DBD:Sc is placed in an otherwise CgPho4 context without the P2ID:Sc (Fig. 6C) and the chimeras were largely non-functional with or without Pho2. Based on these results, we suggested that DBD:Sc requires P2ID:Sc and Pho2 to function in the *in vivo* regulatory environment. Note that our readout in this case was gene induction, not binding. This is contrary to the common belief that DBDs can function independently, which appears true at the binding level but not at the gene regulatory level for ScPho4.

5. One of the problems when making chimeras by swapping IDRs of different lengths, is that the separation between functional elements is now perturbed. It is usual in the literature to try to minimize this confounding. It's difficult here to assess whether this has played a role in some of the 'surprising' results. For example, but the NLS and the P2ID are very large in CgPho4 and splits conserved elements into different spatial locations. One way this has been studied in the literature is to scrambled versions of the non-conserved elements, swap the conserved elements, and test to see if the distance matters. Of course, none of this may solve the potential fact that the breakpoints for making chimeras might not be correct. It seems the breakpoints were chosen to be well within conserved mini-blocks, which I find to be an odd choice.

We appreciate the reviewer's comments and concerns. In our response to a previous comment, we now explicitly tested for the hypothesis that the lack of autoinhibition in CgPho4 was due to the length of its P2ID causing a larger separation between the putative inhibitory element and the DBD. We found this not to be the case. As for the other IDRs with major effects, namely R1 and R3, we separately showed that the functional elements within the CgPho4 regions conferred enhanced activation potential (Fig. 3). We agree with the reviewer that genetic dissection of IDR effects is not straightforward. One of the ways we are moving forward is to exploit natural variation among Pho4 orthologs, which vary in both IDR properties (length and predicted activation potential, for example) and their dependence on Pho2.

As for the choice of the breakpoints, our rationale was to keep intact the known or predicted functional domains and secondary structures. Our strategy was to choose the breakpoints on the edge of well-aligned blocks that correspond to such domains or secondary structure regions. We realize that our original description “we chose the breakpoints in well-aligned regions” can be confusing. We have now revised this to be the language above. Besides, we also tested alternative breakpoints, e.g., for region 3-4 (NLS-P2ID) and found the results to be consistent with our main chimera set. The alternative breakpoint data can be found on our shinyapp <https://binhe-lab.shinyapps.io/Pho4-chimera-data-plotter-v3/> by checking the box “Other”.

Finally, it is usual in this field to interpret positive results of chimeras, but be less assertive with negative results because it is never clear if a negative result is due to the design itself. This is why adding a tiny IDR to the DBD seems to completely change results and so one should be very careful making negative assertions.

We thank the reviewer for the comment. We agree with the general caution the reviewer suggested, but we weren't quite sure which exact example the reviewer was referring to. For the constructs shown in Fig. 6C that were non-functional (hence “negative”), they all had the DBD:Sc with P2ID:Cg. We tested alternative breakpoints for P2ID-DBD to make sure that the result wasn't due to a poor choice of the breakpoints, and all alternative designs resulted in non-functional chimeras (main breakpoints: ScPho4 242-243/CgPho4 458-459; one set of alternative breakpoints were: ScPho4 250-251/CgPho4 463-464). These results can be viewed by using “XXXXCS” as the pattern and check the “Other” checkbox in our shinyapp (<https://binhe-lab.shinyapps.io/Pho4-chimera-data-plotter-v3/>). This led us to hypothesize that DBD:Sc requires P2ID:Sc to function. We then tested and proved this by reintroducing P2ID:Sc into the construct (Fig. 6C). These rescued chimeras are only functional when Pho2 is present, leading us to conclude that DBD:Sc depends on both P2ID:Sc and Pho2 for its *in vivo* regulatory activity in the full TF context.

6. One of the possibly puzzling result from this example is that there are some transcripts that DO require CgPho2 for binding. In He et al (2017), the authors showed that ~14 CgPho4 ChIP peaks require CgPho2. What makes them special? Are the recognition sites different? I would recommend the authors use those promoters as a control for their study here rather than use a single promoter.

We appreciate the reviewer's comment. As the reviewer pointed out, CgPho4 in *C. glabrata* genome exhibited dependence on Pho2 for a subset of its target genes (He et al. 2017, Fig. 5A and associated data). It is important to note that the present study is performed entirely in the *S. cerevisiae* background to keep the *cis* targets and cofactors constant for the comparisons. Thus, the Pho2-dependence states and the associated promoter properties in the *C. glabrata* genome do not directly affect or assist the interpretation of the results here.

That being said, we tested the hypothesis that the 14 Pho2-dependent Pho4 targets in *C. glabrata* have a higher level of nucleosome occupancy, making them “harder” to activate and hence requiring the help of Pho2 (note that our prior ChIP-seq result showed that CgPho2 bound to almost all the sites bound by CgPho4, although that doesn’t imply physical interactions between the two TFs, which our Y2H data didn’t support). We curated publicly available nucleosome occupancy data for *C. glabrata* under rich media conditions, which is appropriate for our inquiry because they show the nucleosome occupancy *before* CgPho4 enters the nucleus. We calculated the nucleosome occupancy at the E-box motifs (when the consensus motif CACGTG was absent, we looked for one-bp-off variants such as CATGTG. ~50% of the sites bound by CgPho4 in *C. glabrata* don’t have a consensus motif). Comparing the nucleosome occupancy between the 14 Pho2-dependent promoters vs the Pho2-independent ones, we found that contrary to our hypothesis, the Pho2-independent promoters had a significantly higher mean nucleosome occupancy level than the Pho2-dependent ones (Fig. R6). We have a few hypotheses to explain this counterintuitive observation, but would need further analyses and experiments to test them.

Fig. R6 Nucleosome occupancy at CgPho4 binding sites differ between Pho2-dependent and Pho2-independent promoters in *C. glabrata*. 14 Pho2-dependent and 86 Pho2-independent Pho4 target genes were obtained from He et al. 2017 Fig. 5 and associated dataset. Published MNase-seq data (Tsankov et al. 2010, PMID: 20625544; SRA: SRR059730, SRR059731) were used to estimate the nucleosome occupancy at Pho4 motifs (CACGTG and all one-bp-off variants) by averaging the per-base-occupancy value calculated by DANPos2 (Chen et al. 2015, PMID: 26301496) and plotted both as dots (per motif) and as a boxplot (box = 25% - 75%; middle line = median or 50%; whiskers = 1.5 times interquartile range). A two-sided student’s t-test comparing the means of the two groups yielded a *p*-value of 0.0062. A total of 41 motifs in Pho2-dependent promoters and 276 motifs in Pho2-independent promoters were analyzed.

As mentioned above, the Pho2-dependent promoters in the *C. glabrata* genome are not suitable controls for experiments conducted in the *S. cerevisiae* background. Instead, we decided to use another well-characterized promoter of *ScPHO84* to validate some of our major results. The reason we selected *PHO84* is because 1) it is highly induced at an early

time point during phosphate starvation, allowing for robust quantification of the chimera activities with and without Pho2; 2) a previous study classified ScPho4 targets into two groups based on their promoter architecture (Lam et al. 2008, PMID: 18418379, Fig. 2). *PHO5* and *PHO84* are the representatives of each group. We therefore believe that *ScPHO84* serves as a good alternative promoter to test the generality of our findings. We transformed the main chimera set into host strains matching the ones we used in the main result except that the *ScPHO5* promoter was replaced by the *ScPHO84* promoter. We found that our major conclusions are supported by the new *ScPHO84pr* results. For example, the new results supported the P2ID as being important for determining Pho2 dependence (Fig. R7A). The split P2ID constructs also support the conclusion that P2ID:Sc encodes two distinct functions: the first part allows for increased activity with Pho2, and the second part inhibits Pho4's function (Fig. R7B).

Here are some minor comments:

1. Line 159, CNNs are 'convolutional' neural networks, not 'convoluted'.
2. Line 294, typo: on 'their' own.
3. The 'nuclear fraction' calculation does not make any sense to me. It does not correct for the size of the nucleus, which can be done by averaging over the area.
4. The proteins are purified with a GST tag, but GST is known to dimerize. Does this interfere with their measurements?

We thank the reviewer for the above suggested edits and questions. We have now corrected both typos listed in #1 and #2.

For #3, the nuclear fraction was estimated by the ratio between the integrated density inside the nucleus and the integrated density in the entire cell. Our rationale for this calculation was that if the NLS from the two parental Pho4 orthologs had different strength, they would result in different nuclear fraction. Upon reading the reviewer's comments, we agree that it is also important to directly compare the nuclear concentration of the Pho4 constructs, which directly relates to the reporter level. Therefore, we now also calculated the average pixel intensity inside the nucleus as an estimate of the Pho4 nuclear concentration. We presented both measures in Fig. S7. Statistical tests showed no difference between the two NLS in either measure. Therefore, our original conclusion is still valid.

For #4, We used the same GST-tagged ScPho4 construct used in a previous PBM study (Zhang et al., 2021, PMID: 33975875) and constructed the GST-tagged CgPho4 in order to use the same antibody and reduce experimental variables. Additionally, Pho4's basic Helix-Loop-Helix DBD functions as a dimer and measures are taken during the size exclusion chromatography step of the purification to obtain protein that corresponds to the dimer size. Therefore, in this specific case, we believe the GST tag is unlikely to interfere with the measurements.